



# High resolution seasonal and decadal inventory of anthropic gas-phase and particle emissions for Argentina

S. Enrique Puliafito[1,2], Tomás R. Bolaño-Ortiz[1,2], Rafael P. Fernandez[2,3,4], Lucas L. Berná[1,5], Romina M. Pascual-Flores[1,2], Josefina Urquiza[1,2], Ana I. López-Noreña[1,2,3], María F. Tames[1,2]

[1] Research Group for Atmospheric and Environmental Studies (GEAA), Mendoza Regional Faculty, National Technological University (FRM-UTN), Mendoza, M5500, Argentina
[2] National Scientific and Technical Research Council (CONICET), Mendoza, M5500, Argentina
[3] School of Natural Sciences, National University of Cuyo (FCEN-UNCuyo), Mendoza, M5501, Argentina
[4] Institute for Interdisciplinary Science (ICB-CONICET), Mendoza, M5501, Argentina
[5] National Agency of Scientific and Technological Promotion (ANPCyT), Buenos Aires, B1675, Argentina

*Correspondence to*: S. Enrique Puliafito (epuliafito@frm.utn.edu.ar)

**Abstract.** This work presents the integration of a gas-phase and particulate atmospheric emission inventory (AEI) for Argentina in high spatial resolution ($0.025° \times 0.025°$; approx. 2.5 km × 2.5 km) considering monthly variability from 1995

to 2020. The new inventory, called GEAA-AEIv3.0M, includes the following activities: energy production, fugitive emissions from oil and gas production, industrial fuel consumption and production, transport -road, maritime and air-, agriculture, livestock production, manufacturing, residential, commercial and biomass + agricultural-waste burning. The following species, grouped by atmospheric reactivity, are considered: i) Greenhouse Gases (GHG): $CO_2$, $CH_4$ and $N_2O$; ii) Ozone Precursors: CO, NOx (NO + $NO_2$) and Non-Methane Volatile Organic Compounds (NMVOC); iii) Acidifying Gases:

$NH_3$ and $SO_2$; and iv) Particulate Matter (PM): $PM_{10}$, $PM_{2.5}$, Total Suspended Particle (TSP) and Black-Carbon (BC). The main objective of the GEAA-AEIv3.0M high-resolution emission inventory is to provide temporal resolved emission maps to support air quality and climate modeling oriented to evaluate pollutant mitigation strategies by local governments. This is of major concern especially in countries where air quality monitoring networks are scarce, and the development of regional and seasonal emissions inventories would result in remarkable improvements in the time + space chemical prediction

achieved by air quality models. Despite distinguishing among different sectoral and activity databases as well as introducing a novel spatial distribution approach based on census radii, our high-resolution GEAA-AEIv3.0M show equivalent national-wide total emissions compared to the Third National Communication of Argentina (TNCA), which compiles annual GHG emissions from 1990 through 2014 (agreement within ± 4%). However, the GEAA-AEIv3.0M includes acidifying gases and PM species not considered in TNCA. Spatial and temporal comparisons were also performed against EDGAR HTAPv5.0

inventory for several pollutants. The agreement was acceptable within less than 30% for most of the pollutants and activities, although a >90% discrepancy was obtained for methane from fuel production and fugitive emissions and >120% for biomass burning. Finally, the updated seasonal series clearly showed the pollution reduction due to the COVID-19 lockdown during the first quarter of year 2020 with respect to same months in previous years. Through an open access data repository, we



present the GEAA-AEIv3.0M inventory, as the largest and more detailed spatial resolution dataset for the Argentine

Republic, which includes monthly gridded emissions for 12 species and 15 sectors between 1995 and 2020. The datasets are available at http://dx.doi.org/10.17632/d6xrhpmzdp.1, under a CC-BY 4 license (Puliafito et al., 2021).

## 1 Introduction

Many political, scientific, and professional efforts are devoted for understanding health and environmental problems. Air quality and global change are certainly two big concerns for present days (Al-Kindi et al., 2020; Haines et al., 2017).

Sophisticated numerical models, chemical transport models (CTM) and general circulation climate models (GCM), are used to identify and proof the underlying physics and chemistry of these environmental and social problems; by predicting the evolution and impact of atmospheric pollutants, as well as their geochemical cycles over space and time. From there on, these models are tools for evaluating and proposing mitigation and reduction strategies to improve air quality (Houghton et al., 2002; IPCC, 2014; Nakicenovic et al., 2000; Ravishankara et al., 2009; Solomon et al., 2009, 2020; Thompson et al.,

45 2019).

Air quality models (AQM) require the association of three types of basic information: meteorological data, static topography and land use data, and spatially gridded emission inventories. Meteorological boundary conditions are usually obtained from local measurements and/or global models such as the ERA Interim (European Reanalysis) and NCEP GFS (National Center for Environmental Prediction — Global Forecast System) reanalysis data. Surface terrain information can be obtained from

satellite data such as those from the Shuttle Radar Topography Mission (SRTM3) (Rodriguez et al., 2005), whereas land use and surface cover data are available from the European Space Agency (ESA) map GLOBCOVER 2009 (Arino et al., 2010; Bontemps et al., 2011) and/or from regional reports (INTA, 2018). Emission data is generally obtained from national or international atmospheric emissions inventories (AEI), which are arranged with different spatial and temporal resolutions, such as, Emissions Database for Global Atmospheric Research (EDGAR) (Crippa et al., 2016; EDGAR, 2019); Evaluating

the Climate and Air Quality Impacts of Short-Lived Pollutants (ECLIPSE) (Stohl et al., 2015); Community Emissions Data System (CEDS) (Hoesly et al., 2018); or the integrated assessment model Greenhouse gas – Air pollution Interactions and Synergies (GAINS) (Amann et al., 2011; Klimont et al., 2017). A comparison among GAINS, CEDS and EDGAR is presented in McDuffie et al, (2020). A review for several national inventories in China is compiled in Li et al. (2017).

Global and regional AEI require a permanent update in the spatial and temporal resolution of their data to keep track of the

local socio-economic developments to improve the results of air quality models and/or global climate applications. Most inventories only present an annual account for a particular year, for example, Huneeus et al., (2020), compares time frame and available resolution of different emissions inventories for countries and cities in South America. National inventories usually include a compilation of Greenhouse Gases (GHG) to comply with international agencies requirements (i.e., UN-International Panel for Climate Change, IPCC). Nevertheless, as these technical reports focus on total nation-wide emissions

for political and governmental protocols, these standard national inventories have low spatial resolution, normally reduced to



a large subnational jurisdiction (i.e, provinces, or districts) and provide low to medium information on activity details. However, good practice in air quality determination and modeling requires the use of the finest possible spatial resolution grid, fine temporal resolution and, whenever possible, technological details of the emissions sectors and activities as well. Gilliland et al. (2003) and De Meij et al. (2006) reported improved modelling results when using high spatial and temporal resolution. The finer the spatio-temporal resolution and the larger the number of species and sectors considered for the emissions, the better the air quality model performance achieved.

Local air quality models use an annual averaged static emissions inventory, whose initial constant primary sources are chemically transported with hourly dynamic meteorological data, resulting in pollution plumes that evolve following the weather conditions. Therefore, implementing a seasonal variable monthly regional emissions inventory, will result in a remarkable improvement in the chemical prediction achieved by air quality models, such as, Weather Research and Forecasting (WRF) model coupled with Chemistry (WRF-Chem) (González et al., 2018; Grell et al., 2005; Ying et al., 2009), CALPUFF (Scire et al., 2000), WRF-CALPUFF (Lee et al., 2014; Tartakovsky et al., 2013); WRF-Chimere (Ferreyra et al., 2016); AERMOD (Cimorelli et al., 2004; Kumar et al., 2006; Rood, 2014). This consideration is important, especially in cities and countries where air quality monitoring networks are scarce, as is the case for most of South American nations, including Argentina.

Atmospheric emission of short-lived climate pollutants (SLCPs), such as $CH_4$, black carbon (BC), CO, Non-Methane Volatile Organic Compounds (NMVOC), NOx ($NO_2$ + NO), $SO_2$ and $NH_3$ affect air quality, ecosystems, agricultural production, and participate in global warming with important radiative effects. In addition, knowledge of the direct emissions of $CO_2$ and $N_2O$ (and the abovementioned $CH_4$) are important due to their dominant role as GHGs within future climate predictions. BC or soot comes from the incomplete combustion of biomass and fossil fuel being a significant constituent of fine particulate matter, an air pollutant associated with premature death and morbidity (Liu et al., 2020; Qin et al., 2019; Segersson et al., 2017). BC has radiative effects by changing the surface albedo when it is deposited or by changing the optical properties of clouds (Myhre et al., 2009; Ramanathan et al., 2001). Methane is an important GHG with high radiative efficiency, it has natural and anthropic sources specially as a component of natural gas, an increasing energy source (Shindell et al., 2004; West et al., 2006). $CH_4$, CO and NOx are precursors of tropospheric ozone, also one of the SLCPs, but since $O_3$ is secondary produced it is usually not included within primary gas inventories (Etminan et al., 2016; UNEP-WMO, 2011). Sulfate aerosols (formed from $SO_2$ and $NH_3$), nitrate aerosols formed from NOx, $NH_3$, and NMVOC emissions have cooling radiative effects (Isaksen et al., 2009). Therefore, reducing SLCPs (except $CH_4$) would produce an improvement in air quality, but would lead to postponing climate change mitigation, requiring some trade-off between air quality and climate change (Arneth et al., 2009). As it is discussed in Stohl et al., (2015), SCLPs emissions, in contrast to long-lived $CO_2$, have different impacts on climate according to its geographic location and time of the year, changing their long-term climatic effect of both GHG and SCLP through multiple interactions (Jacob and Winner, 2009; Shindell, 2015). Thus, detailed spatial and temporal AEIs will help to improve the understanding of these regional and global interdependences.





At the local and regional scale, the detail of temporal and spatial knowledge of the activity included in an AEI will determine the quality of AQM result. For example, the particulate material emitted by a thermal power plant generating electricity will depend not only on the fuel (natural gas, gas oil, or coal), but also on the given generation technology (combined cycle, turbo steam, etc.). Similarly, the increasing use of nitrogen fertilizers in agriculture in Argentina in last 20 years, has allowed the expansion of the agricultural frontier, increasing yields and cereal production, but at the same time, increasing the emissions

of nitrous oxide and ammonia leading to higher SCLP emissions. In consequence, more accurate AEI will contribute evaluating the most efficient measures to reduce SCLP emissions and to assess the economic and health impact of each activity.

This article presents a gridded emissions inventory for a dozen of SCLPs and GHG species in Argentina with high spatial resolution ($0.025° \times 0.025°$; approx. 2.5 km × 2.5 km) and, for the first time, a monthly temporal resolution from 1995 to

2020, including many sectorial activity details compiled in several appendices. It is also a revised extended update and compendium of previously published emission inventories by Puliafito et al, (2015, 2017, 2020b, 2020a) for the years 2014 and 2016, but incorporating additional detailed activities of the manufacturing sector and the monthly temporal evolution for most of the activities and sectors considered (Table A1, App.).We will refer to this inventory as "GEAA-AEIv3.0M": GEAA Argentine High-Resolution Inventory version 3.0 with monthly resolution". We compare our results with the Argentine

GHG inventory for the Third National Communication of Argentina to the IPCC (TCNA, 2015), which includes annual GHG emissions from 1990 through 2014. Annual and spatial distributions of air quality pollutant such as PM and NOx are also compared to the estimations presented in the EDGAR HTAP v5.0 inventory (Crippa et al., 2016, 2020; EDGAR, 2019).

## 2 Material and methods

This section describes the process of preparing the GEAA-AEIv3.0M inventory: how the data from the different activities

were collected, their sources and references, the methodological procedure used to estimate the emissions to the atmosphere, and how the geographical allocation of each activity was performed. Details of each sector are presented in the appendices and supplementary material, providing only representative tables and figures in the main text. Table 1a shows all sectors and activities included in the GEAA-AEIv3.0M inventory, its corresponding IPCC2006 code, the subsections where it is described, and its geographical and temporal extension. Table 1b indicates all species included for each activity with their

spatial and temporal resolution. Table 2 summarizes the names of national agencies and institutions whose activity data was considered here, as well as a compendium of the main acronyms used throughout the text.

### 2.1 Study area and reshaping of databases

The inventory is focused on the activities performed on the continental territory and close coastal maritime area of the Argentine Republic (Figure 1a). Argentina is placed in the extreme south of South America covering 2,778,000 km$^2$ (IGN,

2020). Its political organization includes 24 Provinces and 524 Departments or Districts, split between rural and urban areas.





Population information is available in high resolution such as localities and census fractions. All pieces of data were organized as a gridded map whose cells have a resolution of 0.025° longitude × 0.025° latitude between 53° to 73° west longitude and between 21° to 55° south latitude. An EPGS4326, WGS84 mapping is used (Figure 1a). Thus, the study area is made up of a regular grid of 1441 × 912 cells corresponding to the continental and coastal maritime sector of Argentina.

Figure 1 also shows the different scales associated to the mapping process of the available information.

Depending on the spatial extent, power plants, industrial sources or refueling gas stations can easily be associated with a geographical point; residential consumption and agricultural production to an area source, whereas transport emissions (roads and railways) are associated with a line with a length that can be in the order of hundreds of meters to thousands of kilometers. For air quality modeling purposes, these different source types were reshaped into a single database in the form

of grid map. The resolution of the base information determines the size of the grid cell (in this case approx. 2.5 km × 2.5 km). Area or line sources can either be included or not in a single cell. When sources sizes were greater than one cell (i.e., consumption or production are known at the District level) a proxy known data was selected to spatially disaggregate that variable (i.e., land use, population, etc.). If the variable was smaller than one cell (e.g., small census radii data in urban areas), all the sources contained in that cell were added together (Figures 1 and 2).

The activity data for each sector was obtained consulting official national organizations and reports (Table 2). These included the Statistics and Census Bureau (INDEC), the Ministry of Energy (MINEN), the Ministry of Agriculture and Livestock (MAyGN), the Animal Health Control Agency (SENASA), and the Ministry of the Environment (MINENV) through the Third National Communication of Argentina (TCNA, 2015) to the IPCC, with the subsequent Biennial Updates (for 2014 and 2016).

Fuel production, processing, sales, and consumption for various sectors are available monthly from 1994 to present from public databases at MINEN. Electricity generation and fuel consumption at power plants are available monthly from 1994 to present at the energy distribution agency (CAMMESA) and the Energy Regulation Agency (ENRE). Industrial production is available mostly monthly since 1990 from the respective industrial chambers (see subsections). Transport data is available from several national transport regulation agencies (CNRT: public transport, navigation, and railroad; ANAC: domestic and

international aviation).

## 2.2 Calculation approach

Depending on the specified detail, emission maps are constructed, in a bottom-up process, gathering activity data (i.e., fuel consumptions, number of vehicles, energy generation, etc.) or top-down approach using national aggregated activities (i.e.,

population, total energy consumption, gross domestic product, etc.) and then applying specific emission factors (EMEP, 2019).

The activity data is organized by sectors with monthly resolution from January 1995 up to December 2019, and for some sectors they include several months in year 2020, according to the available information. The general methodology applied is based on European regulations that are compiled in the European Monitoring and Evaluation Program (EMEP) (EMEP,





2013, 2019), and has been described elsewhere (Puliafito et al., 2015, 2017, 2020b). Briefly, emissions are calculated following the general Eq. (1).

$$E(p) = \Sigma[A(i,j) * EF(i,j,p)] \tag{1}$$

where $E$ is the total emission (i.e., Mg/year) for a pollutant $p$; $A$ is the activity of sector $i$, for technology $j$; and $EF(i,j,p)$ is

the emission factor for that sector, technology, and pollutant. For example, the emissions (Mg/year) of CO ($p$), corresponding to the annual consumption of gasoline ($j$), of the private automotive sector ($i$).

The inventory was calculated by each individual sector based on the following steps: first, identifying the source of the emission in its geographical coordinates (latitude and longitude); second, assigning the specific activity that contribute to this emission to each coordinate; third, developing a consistent monthly activity evolution; fourth, applying specific emissions

factors for each species, source and activity; fifth, organizing the information into a three-dimensional map (lat., long., time); and sixth, developing indices, tables, figures, and statistics.

As mentioned above, air quality models (i.e., WRF-Chem) requires fine spatial and temporal resolution (i.e., hourly information); however, the available original activity data is organized in most cases monthly. To obtain weekly and hourly profiles, whenever possible, we evaluated the temporality of each sectorial activity independently. For example, hourly and

daily electricity consumption is available from energy distribution agencies, also the evolution of road transport in large cities are well known. This information allows us to produce an averaged interpolated hourly emission profile, which can later be used as proxy for other sectors (i.e., use of natural gas for heating and cooking). Conversely, other sectors such as agriculture and livestock breeding are only available on an annual basis, and only lineal interpolation may be done to obtain monthly values. Similarly, sectorial information is spatially organized into districts. So, especial care must be taken to

discriminate each information into the merged gridded map. In the next methodological subsections, details are given for the spatial and temporal re-assignation.

## 2.3 Anthropic Emission by Activity Sector

The calculation methodology for each subsector and activity is briefly described below. The data supporting the activity for each subsector, (i.e., monthly fuel consumption, household, technology, number of livestock, etc.), and other relevant

information, was compiled and made available in an external repository as is described in Data availability section.

### 2.3.1 Electricity production sector

The activity and consumption of the electric thermal power plants (TPP) are registered monthly in the Ministry of Energy (Minem, 2020) and in the electric distribution agency (Cammesa, 2020). The location of each power plant is well known, thus in a GIS format, these sources are represented as point sources (Figure 2a). Power plant information included the

available machines and technologies, (CC: Combined cycle, TV: Turbo steam, TG: Turbo gas, DI: Diesel Engine) and the



respective fuel consumption for each machine (NG: natural gas, FO: Fuel-Oil; GO: Gas oil, CM: mineral coal and BD: Biodiesel) (Figure 3a. The emission of each machine and plant is calculated according to Eq. (1), using the proper emission factors.

### 2.3.2 Fuel production sector

Emissions from the production and transformation of fuels were calculated from own consumption, venting, and flaring in refineries, and the production from oil and gas in wells. The Ministry of Energy (Minem, 2020) maintains a monthly record of up-stream (production and extraction of gas and oil) in the wells and down-stream (fuel production, own consumption, and sales) in the refineries. Emissions were calculated from own consumption (in wells and refineries) according to the type of fuel consumed, using Eq. (1). In a GIS format, each well or refinery are represented as point sources, so the emissions are
in their respective coordinate.

### 2.3.3 Transport sector

Emissions can be calculated by applying general emission factors by type of fuel and type of commercialization (Eq. (1)) (EMEP, 2019) for a top-down national total account. However, an inventory dedicated to AQM requires the spatial (and temporal) allocation of consumption activity and emissions. We used a bottom-up approach using GIS software: where roads
and railroads are represented by segments, airports, and navigation ports by points. Activity and emissions are first allocated in the respective segments, and then integrated in the respective grids, as described below.

Road transport fuel consumption for each district (Figure 2c) is available monthly for each type of fuel (gasoline, gas oil, natural gas, kerosene, and liquefied petroleum gas); and by type of commercialization (sale to the public, public transport,
cargo transportation, and agricultural machinery) (data available at MINEM database, Table 3). Additionally, monthly fuel sales are also available for each refuelling gas station (RGS). Thus, we use the location and fuel sales of each commercial RGS (Figure 2d) to estimate the spatial and temporal road transport activity. Road transport fuel consumption is directly proportional to vehicle kilometres travelled (VKT) on each route. The routes are represented as segments on a GIS-type map (Figure A1, App.). These segments intersect the reference grid map (with resolution cells of $0.025°$ longitude $\times$ $0.025°$
latitude). Thus, in each cell there will be small segments that represent the route sections with their respective lengths and hierarchies. The spatial distribution of fuel consumption was carried out following (Puliafito et al., 2015) which synthetically consists of distributing the consumption of each RGS ($Fuel_{RGS}$) using a Gaussian function of variable width (Eq. (2)), according to the type of fuel, and location of the RGS (rural or urban). Then, applying a convolution (Eq. (3)) to calculate in each cell of the gridded map the contribution of each RGS.

$$bg(x,y) = \ \exp\left[-\left(\frac{x-x_m}{d}\right)^2\right] \times \exp\left[-\left(\frac{y-y_m}{d}\right)^2\right] \qquad (2)$$



$$Fuel_{CONV}(x, y, k) = \frac{1}{\sum_{u,v} bg(u,v)} \iint [Fuel_{RGS}(u, v, k) \times bg(x - u, y - v)] du dv \qquad (3)$$

The estimated fuel consumption of each cell ($Fuel_{CONV}$) is distributed proportional to the hierarchy of the routes (highways, main routes, residential and rural roads, etc.). Once the fuel consumption per cell has been obtained, the allocation of the VKT will depend on the fuel efficiency by vehicle type and fuel and the length of each segment in the cell (Eq. (3) and (4)).

$$VKT_{GRID} = R(c, k) \times Fuel_{CONV}(k) \qquad (4)$$
$$VKT_{GRID} = \sum_{k=1}^{K} \sum_{j=1}^{J} \sum_{i=1}^{I} h(j) \times l(i, j) \times veh(i, c, k) \qquad (5)$$

Fuel efficiency is calculated at national and provincial level, according to the balance of fuel consumption and quantity and type of vehicles. Since hierarchy and length are known for each segment, then it is possible to calculate form Eq. (5) the number of vehicles per segment. Finally, the emission can be calculated using VKT and proper emission factors (Eq. (6)).

$$E_{GRID}(p) = VKT_{GRID}(c, k) \times EFc(c, k, p) = \sum_{k=1}^{K} \sum_{j=1}^{J} \sum_{i=1}^{I} veh(i, c, k) \times l(i, j) \times EFv(c, k, p) \qquad (6)$$

Where $EFc$ (c, k, p) is the emission factor for fuel burning (g / to m$^3$ of fuel consumed) and $EFv$ (c, k, p) is the emission factor of each type of vehicle per kilometer traveled (g/km) according to EMEP (2019). Figure 2c shows the fuel sales at the district level; Figure 2d the distribution of the fuel sales for each refueling gas station (RGS); Figure A1 (App.) shows the calculated VKT for gasoline vehicles and the CO emissions, which is proportional to the VKT. This procedure (Eq. (2) to (5)) is then iterated comparing the estimated vehicle flows with those counted by road maintenance agencies. Changes in the hierarchy weights ($h$ in Eq. 5) or gaussian function width ($d$ in Eq. 2) were used to produce the convergence (Puliafito et al., 2015).

Emissions from the domestic aviation sector are estimated based on the landing and take-off (LTO) activity (up to 390 m, or 1000 ft height) and the fuel consumption for cruise phase. (Figure A2e, App.) show the fuel consumption at Argentine airports.

LTO emissions ($E_{LTO}$) and cruise phase emissions ($E_{FLT}$) were calculated following EMEP (2019).

$$E_{LTO}(p, a) = \sum_{k,t} N_{LTO}(a, k, t) \times EF_{LTO}(k, p) \qquad (7)$$

Emissions during the cruise phase were calculated as the difference between total fuel consumption ($E_{FUEL}$) minus LTO emissions

$$E_{FLT}(p) = E_{FUEL} - \sum_{a} E_{LTO}(p, a) \qquad (8)$$



Being $k$, type of aircraft, and $p$ pollutant, $N$: number of LTOs by type of aircraft and $a$: airport in GIS format, the LTO emission were allocated over several cells over each airport according to the orientation of the runways. Cruise emissions were spatially allocated linking airports and frequencies, however for AQM these emissions are not considered since they are emitted at 9000-10000 m.

265

The activity data for the railway park were taken from the National Transportation Commission (CNRT) (CNRT, 2020). Fuel consumption was distributed proportionally to the length of the active railways by applying a hierarchy system distinguishing between full-operating and intermittent rail corridors. Figure A3 (App.) show the railroad (RR) network and the monthly freight and passenger activity. The railroad passenger activity in Argentina is based on a train system based in the city of Buenos Aires that comprises a long-distance service and commuter trains. Many suburban railways lines use electric traction; therefore, their respective emissions are considered in the electricity generation sector. The suburban diesel passenger railways were calculated using the transported passenger-km (PKT), the length of the tracks (LRR) commonly used and the appropriate emission factor for that type of machine.

$$E_{GRID-PR}(p) = PKT_{GRID} \times LRR \times EF_{RR}(p) \tag{9}$$

The railroad freight network is organized to export the production of grains and minerals through the fluvial ports along the main rivers, mainly at Rosario Santa Fe, Buenos Aires, and the deep-water port in Bahía Blanca. In this case, the monthly cargo movement (ton-km traveled TKT) and the fuel consumption of this subsector are known. Emissions were calculated from fuel consumption data and typical emission factors.

$$E_{GRID-RR}(p) = TKT_{GRID} \times LRR \times EF_{RR}(p) \tag{10}$$

Using GIS software, the consumption and emission of each railway subsector and company (freight, passenger, suburban rails) was allocated on segments and then integrated in their respective grid map.

Navigation subsector includes the exhaust emissions from propulsion and auxiliary engines during berthing, maneuvering in harbor and during cruise from ocean-going, in port, and inland waterway vessels. Domestic navigation in Argentina is centralized in the De La Plata, Paraná, Paraguay, and Uruguay rivers. Main active ports are Buenos Aires, La Plata, Rosario, Santa Fe, Campana, San Nicolás, Goya, Reconquista, Barranqueras, Formosa, Gualeguaychú, and Concepción del Uruguay (Figure A3 App.). A general top-down approach was employed to estimate navigation emissions, using available statistics on fuel consumption for national and international navigation, according to the general Eq. (1). Port berths and routes to and from those berths were spatially identified using existing geographic definitions of the port boundaries. GIS tools were used to describe the transit routes using navigational charts. National Port Authority (SSPYVN, 2020) provided the activity data





on every port. Cruise emissions were spatially allocated proportionally across the major shipping lines also using ship movements.

### 2.3.4 Residential, commercial, and governmental sector

The main residential fuel used for heating and cooking in urban centers is natural gas, whose consumption is known monthly for each Province. To spatially distribute this consumption, we used information of household census and a map of census

fractions from the National Statistic Office of Argentina (INDEC, 2020). This map indicates the number of households and population composition in very fine resolution for cities and broader for rural areas (Figure 1c and Figure 1d). We complemented this data with information on unsatisfied basic needs (UBN) to include differences in consumption by households (Puliafito et al., 2017).

$$Rg(x, y, k) = (Hg(x, y, k) * Rd(x, y, k))/Hd(x, y, k) \qquad (10)$$

$Rg$ being the residential consumption of fuel $k$ considered in cell $(x,y)$; $Hg$ is the number of households in the same cell which consume fuel $k$; $Hd$ is the total number of households in district $d$, and $Rd$ is the consumption of fuel $k$ in district $d$. This disaggregation was performed for each type of fuel used for cooking and heating.

In less proportion, especially on rural areas, other heating and cooking fuels are used like wood, coal, and biomass. We assumed a consumption rate for cooking and heating per household of 2.7 Mg (dry basis) for those households which only use biomass, and of 0.25 Mg for the rest of the households (i.e., FAO/WISDOM project in Trossero et al., (2009)). The emissions from domestic use of fuel in each cell are calculated as follows:

$$E_{RESID}(x, y, p) = \sum_k Rg(x, y, k) \times F_{FUEL}(k, p) \qquad (11)$$

where $E_{RESID}(x, y, p)$ are the emissions of pollutant $p$, at cell grid (x, y) resulting from the use of fuel consumption $k$; and $F_{FUEL}(k, p)$ are proper emission factors for pollutant p and fuel type $k$. The emission factors from burning considered are those established by EMEP/EEA (EMEP, 2016) for natural gas stoves and heaters.

Emissions from the commercial sector (small workshops, markets, shopping centers) and government/public office sector (public buildings such as schools and hospitals) were associated with residential emissions. These specific consumptions are obtained from the classification of users of natural gas, the main fuel used that produces local emissions. Note that emissions from electricity consumption in the residential, commercial and government sectors are included in the electricity production sector.



### 2.3.5 Industrial sector

Emissions from the industrial sector were divided into two groups, emissions from in situ fuel combustion and emissions from the production process itself. The consumption of electrical energy from the electrical network is considered in the electricity production sector. Emissions from small manufacturing activities, which do not have significant point emissions to the atmosphere, were included as area sources in the commercial sector.

42 sectors with production-specific emissions were included, identifying more than 450 companies with their spatial location (Figure 2b). Production activity was obtained from the professional chambers of each subsector. These included the following subsector: chemical, petrochemical, refineries, food (sugar, beverages, poultry), non-metallic mining (lime, cement, glass), metallic minerals (iron, steel, aluminum), paper and cellulose (Table A2, App). Regarding fuel consumption, natural gas consumption is known by type of industry and province, for other fuels (bagasse, coal, or diesel) it was estimated from the national energy balance (Minem, 2020). Based on this information, the consumption was set proportional to the production and number of companies in each subsector and province. Electricity and natural gas consumption, and production are known for each subsector, this information was used as proxies to distribute monthly consumption at each company. For the calculation of emissions from fuel consumption, the general Eq. (1) was applied. For the emissions of each subsector, we used the emission factors proposed by EMEP (2019) or EPA AP-42 (EPA, 2016).

### 2.3.6 Livestock and agriculture sector

The inventory of agricultural and livestock activities in Argentina was presented in Puliafito et al. (2020a, 2020b) which considered only 2016 data. An ammonia inventory of Argentina for this sector was presented by Castesana et al. (2018). In this work we extended the 2016 inventory, considering the production of livestock and agricultural activity from 1995 to 2020. To prepare this inventory, we considered the location of livestock raising, the cereal production and the use of fertilizers (Figure 4a and Figure 4c). Animal production is known annually, by type, age of the animal, and production district. The geographical distribution was made proportional to the number of productive establishments (ranches or dairy farms) by department. The emission factors depend on the type and age of the animal and the productive zone.

The production of cereals and other crops is known also annually, by type of crop within each department. The annual quantity of used fertilizers is also known by type of crop. The spatial distribution of the cultivated hectares by type of crop was made using a land use map, distributing in each department the cultivated area and type of crop in agricultural available land. The monthly emissions were simply estimated as proportional to 1/12 of the annual value since the monthly distribution was not available.

### 2.3.7 Burning of agricultural residues and open fires

For the location of biomass burning, crop residues burning, and other biomass fires (natural and / or man-made), we used the MCD64 collection C6 of the MODerate resolution Imaging Spectroradiometer (MODIS) sensor, aboard the (MOD14) Terra



and (MYD14) Aqua satellites (Giglio et al., 2009, 2013), between 2001 and 2020. From years 1995 to 2000 we used information from national fire statistics (Environmental Ministry https://www.argentina.gob.ar/ambiente/fuego/alertatemprana/reportediario and CONAE https://www.argentina.gob.ar/ciencia/conae/aplicaciones-de-la-informacion-satelital/incendios). The MODIS collection

provides two types of products: fire points (fire-events at a daily basis) and burned area (monthly averages, with percentages corresponding to different land uses). The emissions were estimated using the appropriate emission factor corresponding to the specific land use class of each burned area (Puliafito et al., 2020b).

## 3 Results

The present inventory is a multi-dimensional database that embraces spatial coordinates, latitude, and longitude, with a spatial resolution of 0.025°x 0.025° (1441 x 921 cells) for the whole continental and maritime Argentine domain; a temporal resolution of 300 months from Jan 1995 to April 2020, 15 activity sectors and 12 pollutants. It is, then, possible to think of multiple ways to organize and show the results. Therefore, in this Section we will only present some representative figures and tables oriented to compare the absolute and relative contribution of each subsector to the total emission of each species,

as well as to highlight the spatial and temporal variability for the whole country and within different regions. Note that the whole database has been published for its use in air quality / climate model applications in a standardized format within a free access repository as indicated in the Data availability statement. Figures 3 to 6 show selected sectors and species distribution. Figures 7 to 9 cover the results of comparing GEAA with other commonly used inventories.

The multiple appendix and supplementary information provides the monthly and annual emission time-series, as well as

basic representative figures.

### 3.1 Electricity production sector

As of December 2019, Argentina had a total installed capacity of 39,704 MW, where 64.3% (25,547 MW) corresponded to sources of thermal origin; 28.5% (11,310 MW) to hydro; 5.3% renewable (2,092 MW: 1609 MW wind, 439 MW solar and 42 MW biogas: 2 MW); and 4.4% (1,755 MW) nuclear. Annual thermal generation reached in 2019 80,137 GWh, hydraulic

35,370 GWh, nuclear, 7,927 GWh, and renewables 7,812 GWh. Figure 2a show the spatial location of thermal power plants in Argentina. Annual thermal generation for 2019 was produced using mostly natural gas (17,209.2 million $m^3$), diesel (403.8 thousand $m^3$), fuel oil (185.6 Gg) and mineral coal (221.8 Gg), with an average efficiency of 1858 kcal/kWh. Figure 3a shows the total energy consumed at TPP according to the type of generation. The GHG emissions variation, in terms of $CO_2$ eq. (GWP100: $CO_2$=1; $CH_4$=25; $N_2O$=298) (Myhre et al., 2013), is shown in Figure 3b and Table 3. The monthly

evolution for several pollutants is shown in Figure A2a (App.). The large variations in these emissions were associated with three important variables. a) A low frequency variation (with a maximum between May 2015 to May 2017 and minimum in Dec. 2002), corresponding to the economic activity that impacts generation and fuel consumption. b) A variation of medium



frequency, corresponding to the seasonal summer / winter variation, which depends on the ambient temperature, with heavy consumption in the summer months, for example, due to the use of air conditioning. c) A third variation of greater frequency

was associated with the type of fuel. An increasing proportion of natural gas use and a decrease in gas oil and coal are shown in Figure S3(Suppl. Mat). These have been reinforced in recent years due to increased natural gas production from the Vaca-Muerta basin (approx. 38.64 °S, 69.86 °W) from non-conventional wells (Minem, 2020; Rystad, 2018). Figure A2b (App.) also shows that during austral winter months TSP emissions (and $SO_2$) increased and those of NOx decreased. This is due to the reduction in the use of natural gas (the main residential heating fuel) and an increase in coal and fuel oil in power plants

to compensate the natural gas reduction. In summertime the opposite occurs, larger use of natural gas and a reduction of fuel oil and coal results in higher NOx and lower TSP. Note that during diurnal high electricity demand (peak hours) the thermal plants may also be covered by fuel oil and gas oil. In terms of GHG, emissions from electricity production have steadily climbed around 2 % per decade, from 7.1 % in 1995 (with respect to total annual -all sectors) towards 11.7 % in 2019. NOx values have increased from 10.2 % to 14.5 % (with respect to total annual -all sectors), during the same period.

**3.2 Fuel production sector**

Emissions from fuel production correspond to own consumption at refineries (ROC), and extraction wells, for their own operation of the activity and transformation (FPR), and fugitive emissions from venting or flaring of surplus gas are also located in refineries and wells (FUG). Figure A2d (App.) shows the monthly variation between the years 1995 to 2020 of methane emissions reaching a monthly average of 27,835 Mg per month for the three activities. However, the total $CH_4$

emission is dominated by the refinery venting and flare activity. The increase after Nov 2018 is mainly due to a growth in the production of unconventional natural gas in the Vaca-Muerta basin in the last two years (Figure A2c; App.). Figures S6 and S7 (Suppl. Mat.) also shows the activity and emissions of the extractive activity of gas and oil (up-stream) at wells from own consumption. Monthly GHG emissions have increased from (ROC+FPR+FUG) 1403.20 Gg $CO_{2eq}$ in Dec. 1995 until reaching 2457.20 Gg in Dec. 2019. Table 3 show the total annual emissions for oil and gas production for all pollutants

considered. Fuel production and transformation (ROC+FPR+FUG) represented 6.8 % in 1995 and 10.3 % in 2019 of total GHG annual considered. Pollutants such as CO and NOx have an annual contribution share of 1.1 % and 4.1 %, respectively, for year 1995 and 1.9 % and 5.1 % for year 2019, respectively (Table 3 and Figure 5).

**3.3 Transport sector**

Figure A1c (App.) shows the monthly country fuel sales variation for main fuels used in the road-transport sector (ROT)

from Jan 1995 to Dec 2019. Figure A1d (App.) presents the total monthly emissions of CO, $NO_x$ and PM10 from the same activity. Table A4 (App.) show a growth of 13% in the period January 1995 to December 2019, for $CO_2$ and $CO_{2eq}$, 54 % methane, 21 % $NO_x$ and 20 % CO and NMVOC for the same period. The main growth is due to the higher consumption of gasoline while diesel oil has only grown slightly and CNG has remained stable. However, similarly to the energy production sector, fuel consumption is strongly linked to economic activity (i.e., represented by the gross domestic product GDP as we



will discuss later in Section 3.7) showing decreasing consumption from 1995 to 2002, and then climbing again. From August 2016 and on, a stagnation in gasoline consumption appears, in accordance with a retraction in national economic activity. Figure A1c and A1d (App.) also show a 52% and 63% reduction in NOx and CO ROT emissions respectively (comparing April 2020 with respect to April 2019), due to the COVID19 quarantine effect (which began on March 20, 2020) Table A5 (App.) (Bolaño-Ortiz et al., 2020). Additionally, Figure S8 (Supp. Mat.) includes monthly and annual GHG emissions ($CO_2$,

$CH_4$ and $N_2O$) and SLCP (BC, CO, NMVOC; NOx, $SO_2$, $NH_3$) from road transport. Regarding domestic aviation (DOA), Figure A2e (App.) show monthly fuel consumption ($m^3$) from LTO, while Figure A2f (App.) show the respective monthly emissions ($CO_2$, $CH_4$, $N_2O$, NOx, CO, NMVOC, $SO_2$ $NH_3$, TSP and PM) The aviation activity has been relatively stable with increasing trend from year 2005. Year 2020 had a complete stop due to COVID-19 restrictions, only partially reestablished after Nov 2020.

Figure A3 (App.) shows the active railroad network (A3a), the average seasonal variability in RR activity (A3b), in terms of t.km for freight and passenger.km for transported passengers; and the monthly fuel consumption and amount of transported passenger (A3c). The passenger activity is centered in Buenos Aires commuting activity (> 95 %). With respect to fuel consumption (gas oil), RR freight activity represents on average 45 % and it is expected to increase as crop production and export increases. Note that following the agriculture exportation, freight RR shows a marked seasonality, where the

maximum austral winter activity (June-July) is up to 40% higher than during the summer (Jan-Feb). The inter-annual increase is also seen in inland navigation since ports like Rosario, Santa Fe and Bahía Blanca are concentrating the soja-bean, wheat, and maize export. Both railroad and inland navigation activity have shown an increase of 122 % in pollutant emissions from Dec 1995 with respect to Dec 2019.

## 3.4 Residential, commercial, and governmental sector

Residential, commercial and government (R+C) energy consumption includes electricity (for lighting, air conditioning and partially heating), and natural gas for cooking and heating in a large part of the country (except for northeast Argentina, see Figure A3, App.). For urban areas not connected to the natural gas (NG) network, the heating energy consumption is replaced by electricity, LPG, kerosene; and in rural areas with abundant biomass available (northeast of the country), the use of charcoal and wood. According to data from radio maps and census fractions, there are 12,171,560 homes in Argentina

(INDEC, 2010), of which 56% are connected to the NG network, 41% use LPG, and the remaining 3%, wood, charcoal, or kerosene. The 2019 annual consumption reached 10,680,070 (1000 $m^3$) of NG, 855,184 (1000 $m^3$) of LPG, 285,113 Mg of wood, 341,473 Mg of kerosene and 484,408 Mg of coal. The annual average per capita consumption is 268 $m^3$ of NG; 21.38 $m^3$ of LPG; 12.11 kg of charcoal, 7.1 kg of firewood, 8.5 kg of kerosene. Figure 3c shows that the annual fuel consumption of wood and LPG has decreased since 2001 and 2007, respectively, compensated by a gradual increase in the consumption

of NG since 1995. Note that the residential fuel consumption shows a very strong seasonal and regional cycle (figures 3d and Figure A3 App.) due to the large North-South extension of Argentine territory. For year 2019, NG uses represent the 80% of the total R+C annual emissions for $CO_2$, 12% for $CH_4$, 91% for NOx, 20 % CO and 10% for TSP; complementary, the use of





other fuels contributes to 91% for $PM_{10}$ and 90% of CO. (Table 3; Table A4 and A5 App.). Emissions from R+C electricity using fossil fuels are considered in the Thermal Power Plant sector.

### 3.5 Industrial sector

This subsection includes the monthly emissions from industrial manufacturing own fuel consumption (MFC) and emission from the production process (MOP) from January 1995 to April 2020. Note that manufacturing electricity consumption is considered in the thermal power plant sector. Table A3 (App.) shows a list of the manufacturing activities considered whereas Figure 2b shows the location of the manufacturer sector. The monthly fuel consumptions averages are: 846,380 (1000 $m^3$) of natural gas, blust-furnace gas, and coke-oven gas together; 13,493 Mg of LPG; 36,234 Mg of gas oil, diesel-oil, and fuel-oil; and 668,374 Mg of coal wood and biomass. Natural gas is used as main own fuel consumption followed by wood and crop residues, the later especially used in the food elaboration subsector, like sugar production, paper and wood manufacturing, due to local availability of biomass. Seasonal fluctuations, both in consumption and emissions, are due to variations in production, but it is also conditioned by less availability of natural gas during the winter months, which is derived to residential consumption. Monthly average GHG from own fuel consumption reached 1667.34 Gg per month of $CO_{2eq}$, while for NOx reached 3743.16 Mg, 1603.43 for CO and 1250.46 Mg for TSP.

The MOP included the emissions from the manufacturing own production process and included the following subsectors: 2A glass production; 2B chemistry, 2C aluminum-steel, 2D asphalt, painting, 2H paper, food, beverage. Figure 3e and Figure 3f shows the annual evolution of MOP NOx and $PM_{10}$ emissions. Chemical industry contributes to 37.1 % of NOx emissions, followed by the food industry 36.5 %, and the steel industry with 26.4 % with respect to total MOP emissions. For $PM_{10}$ emissions, the cement industry contributes 35.0 %, the chemical industry contributes 22.2 %, followed by the steel industry with 20.6 %, and the food industry 20.4 %, automotive painting contributes with 1.8 %.

### 3.6 Agricultural and livestock sector

Emissions from the agricultural livestock sector were calculated annually from 1990 to 2019. Emissions from livestock included enteric fermentation ($CH_4$) and manure management ($CH_4$, $NO_2$, $NH_3$, NOx, NMVOC, and PM). These emissions depend on the type of animal, age, type of production and the productive areas. In terms of methane emissions (i.e., CO2eq), the bovine sector dominates Argentina's GHG emissions (30.9 %), reaching 95,473 Gg $CO_{2eq}$ in 2019; (2781.09 Gg $CH_4$; 87.09 Gg $N_2O$). The historical series shows an average of 96,301 Gg $CO_{2eq}$ between 1995 and 2019 for all livestock production (Figure 4b) with a slight decrease in 2009 by a reduction in bovine animal production. Total animal production has grown from 177 million head in 1990 to 317 million head in 2019. While bovine livestock has oscillated between 54.7 ± 3.4 million head, the largest increase was shown by the poultry sector rising its production of 30 million birds in 1990 to 232.3 million in 2019 producing a significant increase in ammonia emissions (from 6.6 Gg $NH_3$ in 1990 to 51.1 Gg in 2019, see Figure. 4a). Total ammonia emissions in 2019 reached 211.63 Gg for all livestock.





Emissions from the agricultural sector are signed by a strong increase in cultivated area, increased production, and increased
use of fertilizers (Figure 4c). Considering the period 1990 to 2019, these numbers more than doubled from 17,700 to 37,873
kHa in cultivated area; approximately tripled from 51,457 to 172,089 Gg for cereals production; and increased at least by a
factor of 15 (from 260 to 4217 Gg) for fertilizers use. Consequent to this increase of fertilizers, the largest emissions
increases were for ammonia and nitrous oxide, which changed from 38.09 Mg in 1990 to 531.58 Mg in 2019 for $NH_3$ and
from 1.58 Mg in 1990 to 22.11 Mg in 2019 for $N_2O$ (Figure 4d).

**3.7 Burning of agricultural residues and fires**

For this sector, accidental and/or provoked fires from biomass burning were considered, both from agricultural residues or
from other types of fires between 1995 and 2020. Figure 4e shows an average seasonal burned areas according to main land
types and Figure 4f shows the evolution of $PM_{2.5}$ (Gg) emissions for the period 1995-2020, according to land type. Figure
A5b (App.) shows the monthly average precipitation (1981-2018), calculated using the Climate Hazards Group Infrared
Precipitations with Stations (CHIRP) database (Funk et al., 2015; Rivera et al., 2018). It clearly shows the correspondence
with the land use map (Figure A5a, App.), and directly with the availability of ground fuel from biomass. Figure A5c (App.)
shows the average monthly burned area (2001-2020), which shows two distinct area: North- east (rain >50 mm/month) and
the semi-arid (rain > 20-50 mm/month) central-west zone of Argentina. In the northeastern area of Argentina fires
predominate between August and November, associated to burning of crop residues and land changes (clearing forest for
agriculture), while in the center-west of the country fire events increases during the summer months (Dec and Jan) on dry
grasslands and pastures. These fires are associated to typical dry conditions in the previous winter and spring months before
the raining seasons begins in late summer (Feb. and March). Figure A5c (App.) shows the emission of $PM_{2.5}$ associated to
burning of biomass.

According to land type use considering the 1995-2020 period, annual burned area averages 1,064,423 million Ha, being 14.7
% forest, 27.1 % grassland, 25.6 % savanna, 22.0 % shrublands, 7.7 % cultivated areas, and the remaining 2.9% corresponds
to other type of land uses.

**3.8 Summary and discussions of GEAA-AEIv3.0M results**

Table 3 summarizes the total annual emissions for years 1995 and 2019, while Table A4 (App.) presents a single timeframe
with the monthly emissions for December 1995 and December 2019. From the point of view of the GHG emissions, the
main emission sector is livestock (41.5 % and 31.7 % for 1995 and 2019, respectively), showing a 9.8% reduction trend due
to decreasing bovine production (see Figure 5). Adding together thermal power plants and manufacturing own fuel
production represents 14.8 % and 18.3 % of the total GHG emissions (for 1995 and 2019, respectively), followed by road
transport 16.9 % and 16.8 % (1995, 2019 respectively). The residential plus commercial sectors have increased from 8.6 %
to 9.7 % for the above referenced years. This is consistent with population increase, as analyzed below. In absolute values
GHG have increased from 244,420.25 Gg $CO_{2eq}$ in 1995 to 301,618.79 Gg $CO_{2eq}$ in 2019 (18.9%). Note that GEAA-





AEIv3.0M GHG inventory does not include land use changes nor sewage waste, since its focused-on air quality, and therefore these are not the total GHG numbers for Argentina; in fact, TCNA (2015) reports total $CO_{2eq}$ of 368.295 Gg for year 2014. Most notably, the main increases are observed for $NH_3$ and $N_2O$ emissions due the use of fertilizers in the agriculture (Figure 4d). Indeed, Argentina has increased its annual crop production from 51,735 to 172,089 Gg, and annual

use of fertilizers from 641 to 4217 Gg (1995, 2019 respectively), while bovine production has decayed slightly from 55,921 in 1995 to 54,698 thousand heads in 2019 (Figure 4a). From a climate change perspective, reducing $N_2O$ emissions through reducing crop production is a critical economic option, since together with livestock feeding, both activities represent the main export income for Argentina. Thus, it is not expected that the percentage contribution of $N_2O$ to Argentine GHGs will be reduced until new nitrogen-uses efficiency of crops could be incorporated worldwide to reduce emissions (Solomon et al.,

2020; UNEP, 2013).

Air quality SCLP sectorial shares are shown in Figure 5b and Figure 5d for 1995 and 2019, respectively (see also Table 3). Comparing those two years for a particular pollutant, e.g., CO, show that the dominant sectors contributing to the total emissions remain unaltered and present only minor percentage changes: road transport is the most important sector representing 76.7% and 82.8% for years 1995 and 2019, respectively, followed by open fires (12.1 % and 5.7 %) and

burning of agricultural residues (2.4 % and 1.8 %, for years 1995 and 2019, respectively). Similarly, NOx, emissions are concentrated in the road transport activity, 58.8 % and 50.8 %; thermal power plants and manufacturing own fuel production, 17.9 % and 19.7 %; as well as residential & commercial, 10.7 % and 9.4 % (years 1995-2019 respectively). Fire and biomass burning represent the largest source of particulate matter (TSP) (47.2 % and 28.4 % for years 1995-2019 respectively) coming from agricultural waste, cleaning forest for agriculture and livestock feeding and natural burning of grassland.

Nevertheless, it should be noted that the TSP contribution from different sectors is highly variable from year to year (Figure 4f).

The three concentrically rings presented in Figure 6 summarize the sectorial contribution to the main primary air-quality pollutants (see also Table 4): the outer ring is for $PM_{10}$, the middle ring for NOx and the inner ring corresponds to CO. Figure 6a show the proportion of total annual emissions with respect to urban population density. 87.5% of total PM10

emissions (231.867 Mg), 63.5 % of total NOx emissions (544,373 Mg) and 60% of total CO (2,331,849 Mg) are emitted in areas with low urban density (< 100 inhab./km²), since many roads and thermal power plants are in these locations and Argentina has a vast non urbanized area (see Table 4). Note that Argentina's populations live 25.9 % in towns with less than 1000 inhab./km²; 69.4% in urban centers between 1000 and 10000 inhab./km² and 4.7% in dense urban centers greater than 10000 inhab./km². Air quality in urban areas is dominated by road transport, residential & commercial emissions, and

depending on the cities also by power plants and industrial energy consumption and production. For example, for NOx, the population is exposed to average daily emissions of: 0.5, 8.6, 35.7, 114.6 and 216.6 kg / km² / day for <=100; >100 and <=1000; >1000 and <=5000; >5000 and <=10000; and > 10000 inhab./km²; respectively. However SCLP high emissions density per squared km is emitted in the denser urban area (>10000 inhab./km²): 1,557 kg/km²/year for PM10, 79,070 kg/km²/year for NOx, and 431,755 kg/km²/year for CO (Figure 6b), resulting in those urban regions to possess lower air





quality standards than rural areas. Figure 6c show the proportion of the same SCLP ($PM_{10}$, NOx and CO) but as function of the sectors. These figures show, that although CO and NOx have the highest emissions density in urban centers and are dominated by road transport and power plant, maximum $PM_{10}$ is located in medium density areas (4,612 kg/km$^2$/year at urban density of >5000 inhab./km$^2$ <=10000) and are dominated by residential and road emissions. Nevertheless, in absolute numbers PM is dominated by AWB and OBB produced in agriculture and forest areas.

The evolution of GHG and SLCP air pollutant emissions clearly show a strong dependence to population increase and gross domestic product (GDP) changes. Figures A6 (App.) show a normalized quarterly series of GDP, de-trended population, and GHG. While population follows a linear trend (0.04% quarterly increase), GDP has a 6–8-year oscillation over the population increases, presenting local minima for Oct-2002 and Apr-2020, and local maxima for years Apr-1998 and Apr-2013. GHG variation follows the GDP changes with an extra annual seasonal variation. Note that the medium term 6–8-year

oscillation as well as the annual seasonality are appreciable in the use of fossil fuels for electricity production, as described in Section 3.2. Finally, Figures A6c (App.) show the GHG/cap and GHG/GDP variations, whose trends are followed by the emission of many other pollutants (not shown). Several conclusions may be extracted from the above results. First, GHG and air quality pollutants follow mainly population increase modulated by economic activity, where Argentina's recurrent economic crisis are very visible in these timeseries. Second, GDP variation during year 2019 has fallen more rapidly than the

population increase during the same period, aggravated by COVID-19 lock down crisis in early 2020 (Bolaño-Ortiz et al, (2020); see Table A5 (App.) for monthly values for April 2019 and April 2020). Third, quarterly GHG/cap has been stable on 671 ± 62 kg/cap during the whole period, which means there have been no major enhancements in personal consumptions, but neither have been any improvement in the emissions efficiency. Fourth, GHG/GDP show a quarterly variability of 54.3 ± 22 g/USD, showing a slight decreasing trend from 2004 to present, since less carbon is emitted per

expended USD, most probably due to technological changes (note that the sudden increase in year 2002 is produced by the reduction of GDP during the 2001-2002 economic crisis). Fifth, approximate one third of GHG emissions comes from agriculture and livestock emissions, main export activities of Argentina, another third arises from energy production (TPP) and transport (ROT+DOA+R+N), and the remaining third from the other sectors. Sixth, GHG are still coupled to GDP (and population), which means that reducing GHG emissions in Argentina can only be done, at present, at the expense of reducing

activity intensity (i.e., reducing economy), as it is clearly seen in year 2020 reduction due to lock down due to COVID-19. Seventh, air pollution in urban cities is mainly produced by road transport (i.e., CO, NOx and $PM_{2.5}$) and power plants ($SO_2$ and NOx), and even though the largest emission densities are in large urban areas, due to the vast majority of rural areas in Argentine territory, the total national emission are originated in the less populated regions.

**4 Inter-comparison of GEAA-AEIv3.0M with other Emissions Inventories for Argentina**

Since the present GEAA-AEIv3.0M inventory includes highly resolved spatial and temporal activity reports, its calibration requires a double control and validation. For the temporal comparison we use the Argentina national greenhouse gas inventory (TCNA, 2015) that compiled the total annual values for Argentina between 1990 and 2014. Although the activity





data for both studies were taken basically from the same national sources, the focus and methodology of each inventory varies. In TCNA activities and emissions are accumulated using a top-down approach to obtain a nation-wide annual total by
sector. While in our case (GEAA-AEIv3.0M) the activities and emissions are first located in each point, line, or area with a bottom-up approach, and then the totals are calculated as the sum of all cells in the spatial grid. Therefore, the sum of the activities by sector and year may vary slightly. Likewise, we compare the annual values with the international EDGAR inventory, which differs especially in the use of proxy variables used for its spatial disaggregation, which has already been discussed elsewhere (Puliafito et al., 2015, 2017). A spatial comparison can also be made with the EDGAR inventory,
although it has a resolution of $0.1° \times 0.1°$, which requires an adaptation of our higher resolution inventory ($0.025° \times 0.025°$). When comparing with other inventories, emphasis has been placed on greenhouse gases (GHG), since GHG relates to the level of agreement (or discrepancy) with the activities of each sector, since their emission factors (EF-GHG) are well established and are especially associated with energy consumption (Sato et al., 2019). On the other hand, air quality emission factors (EF-AQ, those used for NOx, CO, PM, and others) are highly variable, mainly due to uncertainties in the
environmental and technological conditions considered for each activity. For example, for an on-road vehicle, the emission factors will depend on the outside temperature, engine temperature, type, and quality of fuel, idle or regime status, slope, load, age, among other factors (EMEP, 2019). So, the used average EF-AQ, will include a mixed weighted operational condition. In the same line, although electric vehicles have EF-AQ = 0, EF-GHG will still depend on how the consumed electrical energy is generated.

## 4.1 Comparison with total annual values from TCNA

Table 5 and Table A6 (App.) summarizes the total annual values for GHG emissions ($CO_{2eq}$ Mg) for GEAA-AEIv3.0M and TCNA inventories, respectively. Note that the original TCNA report included contributions from other sectors (land use changes) not related to air quality that are not considered here. Figure 7a shows the annual values for both inventories, and Figure 7b shows the average annual differences by activity Table A7 (App.). Most of the activities (1A1, 1A2, 1A1b, 1A1c,
1A3a, 1A3b, 1A4a-b, 2B, 2C, 3A, 3C, see Table 1a) agree within ± 6.0 % with total differences for the sum of all sectors of 0.4 ± 3.9 %. Higher discrepancies are found in sector 1A1c (FPR 7%), 1A3c-d (R+N: 13.3%), 3C (AG: -12.5%) and (AWB - 6.5%). For fuel production, the discrepancy arises from the way the activity is computed. In TCNA this is estimated from total oil and gas production, while GEAA computes emissions from own consumption at wells. Railroad and navigation activity have been updated recently (posterior to the TCNA publication) and therefore there might be differences in the
estimated activities, mostly in the consumption of natural gas for domestic transport. For agriculture, and biomass burning the total differences between inventories arise from the activity data: TCNA uses national accounted fire statistics, while in GEAA, MODIS MDC64 data were used for years 2001-2020 (see Section 2.3.8). Figure A6 (App.) show annual GHG emissions comparison for the energy sector excluding refining and fugitive emissions from fuel production, resulting in a very good agreement between GEAA, TCNA and EDGAR for the main energy sector when the same aggregation scheme is
applied. EDGAR however has 2.5 times more $CH_4$ emissions for the fuel production sectors (1A1bc,1B1,1B2) than GEAA



and TCNA (see discussion below). The methane emissions from fuel production and fugitive emissions from oil and gas well needs a deeper study since a bottom-up calculation from each possible source requires in situ / airborne measurements to detect possible leakages from local facilities (Allen et al., 2013; Roscioli et al., 2015; Zavala-Araiza et al., 2014). New high-resolution satellites promise new detection capabilities (i.e., GHGSat. https://www.ghgsat.com/our-platforms/iris/).

**4.2 Comparison with EDGAR database**

Spatial and total annual emissions were compared to the EDGAR emissions inventory (EDGAR HTAP v5.0) for Argentina. In particular, the EDGAR monthly inventory is available only for year 2015 (Crippa et al., 2020), which was used to compare the GEAA-AEIv3.0M monthly values. Table A8 (App.) shows a summary of the statistics obtained from this comparison. For this purpose, The GEAA-AEIv3.0M inventory was adapted from a 0.025° to 0.1° spatial resolution compatible with EDGAR.

Figure 8 shows the annual spatial differences between both inventories for PM10 for the transport sector (panel a), for the residential and commercial sector (panel b), as well as the annual total evolution for both sectors (Figures 8c and 8d, respectively). Figure 9 shows equivalent information as Figure 8 but for NOx.

The GEAA-AEIv3.0M vs. EDGAR HTAP v5.0 comparison shows several interesting aspects. From the spatial point of view, the residential emissions shown by EDGAR has a distribution based on the districts with surface emissions larger than the properly urbanized area, see for example, green-blue areas in north-west Argentina (Figure 8b for PM$_{10}$) which corresponds to a mountainous and arid area, with practically no population and only minor industry based on agricultural waste burning. The larger EDGAR emissions (negative values) for the whole district are clearly an overestimation due to not considering a high-resolution population density map, as there are no direct sources on most of this region, and most of the emissions are located on a unique location on the east-edge of the district (see red dot). When appreciating the annual values, the differences of PM$_{10}$ (and other pollutants), show similar values between the years 1995-2000, but thereafter diverges. This difference arises from a possible overestimation on the EDGAR inventory on the amount of firewood and charcoal used for heating and cooking in homes. In effect, this amount has been decreasing significantly since 2002, being replaced by an increase in the use of natural gas and LPG (Figure 3c); therefore, EDGAR trends should be corrected (Figure 8d). For estimating the residential emissions, as mention in Section 2.3.4 GEAA uses the census fractions map (INDEC, 2020) which gives fine detail of the location and amount of homes, specifying the main fuel used for cooking and heating (natural gas, wood, etc.). For NOx (Figure 9d) EDGAR overestimates GEAA values, which is seen as mostly blue areas (negative values) in Figure 9b. Since both annual series show equivalent variations, it is most probably that the discrepancies arise from the use of different emissions factors in each inventory.

Regarding transport emissions, the spatial distribution differs in the amount of traffic and emissions per route. On the EDGAR map equivalent emissions have been attributed to primary and secondary routes (see light blue lines in Figure 8b), whereas the GEAA-AEIv3.0M distinguished among routes hierarchy (see red lines in Figure 8b). Although the annual total emissions are similar, this oversizing produces less emissions on main routes for EDGAR. It should be considered that



national freight transportation by trucks in Argentina (95 % of land freights) is more important than freight transportation by
trains or ships.

Table A8 (App.) show the following aspects: On the one hand, emissions from fixed sources, thermal power plants and
industries) have a very similar representation between inventories (< 25 % relative difference) and little variance, which
indicates that the activity is similar but with a slight difference in the used emission factors. On the other hand, for the fuel
production and fugitive emissions subsectors (1A1cb, 1B1 and 1B2), GEAA-AEIv3.0M has an important difference with
respect to EDGAR, especially in methane emissions being EDGAR more than 90 % larger than GEAA (for the sum of
subsectors). These differences totalize 598 Gg of $CH_4$ (or 14,970 Gg $CO_{2eq}$) per year (Figure 7 and Table 8 App.). Note that
for the particular case of  the 1B1 sector (fugitive emissions from coal mining), the activity data for the GEAA inventory has
been estimated from the national primary energy balance, which possess large uncertainties (TCNA, 2015). Agriculture also
shows important differences (> 150%) for nitrous oxide. These differences arise from direct and indirect emissions of $N_2O$ in
manure management and managed soil, but as GEAA does not include land changes, our emissions might have been
underestimated in comparison to EDGAR. Estimation of biomass burning activity (AWB, OBB) also has large uncertainties
in determining burned crop residues and land fires, resulting in relative emissions differences > 120% between GEAA and
EDGAR. In contrast, average $CH_4$ emissions have relative difference of less than 70 % for most of the sectors. Similarly, for
most of SCLPs, differences range between 5 % and 65 %, with a general lower estimation of pollutants emissions for
GEAA-AEIv3.0M with respect to EDGAR.

**4 Conclusions**

A multidimensional inventory of emissions of air pollutants to the atmosphere of Argentina for 15 activities and 12 species
has been compiled. This new inventory has a monthly temporal resolution (300 months between 1995 and 2020), and a high
spatial resolution of $0.025° \times 0.025°$. The activities included are energy production, fugitive emissions from oil and gas
production, industrial own energy and production, transport -road, maritime and air-, agriculture, livestock production,
residential, commercial and biomass burning. Twelve species were considered: GHG $CO_2$, $CH_4$, $N_2O$; ozone precursors: CO,
NOx, NMVOC; acidifying gases: $NH_3$, $SO_2$; and particulate matter: $PM_{10}$, $PM_{2.5}$, TSP, BC.

The main objective of the emission maps is to support air quality and climate modeling, as well as to evaluate in time and
space pollutant mitigation strategies. In fact, the calculated pollutant temporal series clearly showed the pollution reduction
due to the COVID-19 lockdown during the first quarter of year 2020 with respect to same months in previous years. This
situation gave us also the opportunity to link the pollutant emissions to economic activity, showing how Argentina's
emissions are still very much coupled to population and GDP, therefore an (expected and needed) economic recovery will
surely increase emissions impoverishing the air quality. In fact, 31.7 % of GHG emissions comes from livestock feeding (in
rural areas), and more than 60% of total SCLP emissions are emitted in rural areas (mainly both from agriculture and
transport), representing altogether the main export activity of Argentina. Note than in general, emissions density is very low
in most of Argentina territory, but SCLP emissions density in middle-size urban areas (pop. density > 5000 inhab./km$^2$) are





very high due transport and power plants. Investments in technology and the promotion of de-carbonized activities for reducing and decoupling GHG, and air pollutants from GDP will require big investments and further fostering cultural changes (i.e., like bicycling in cities, changes in public transportation), which will still take many years. As it has been noted in the electricity generation, thermal power plants operate mainly with natural gas, but needs to use gas oil or coal during peak hours and in winter months, therefore, air quality improvement has less room in this sector than it could be achieved in the urban road sector (i.e, electric motorization).

Finally, we compared the GEAA-AEIv3.0M results against the Argentine GHG inventory of the Third National Communication of Argentina to the IPCC (TCNA, 2015), which compiles total country wide annual GHG emissions from 1990 through 2014, agreeing within $\pm$ 4%. Spatially and temporal comparison was also done with EDGAR HTAPv5.0 inventory for several pollutants. The agreement was acceptable within less than 30% for most of the pollutants and activities, although a discrepancy bigger than 90% was obtained for $CH_4$ arising from fuel production and > 120% for biomass burning. Note that $CH_4$ emissions from fuel production are a permanent concern due to its big greenhouse potential effect, therefore more detailed studies will be required to unravel the differences, since top-down inventories requires a great effort to assess the actual emission chain.

Seasonal variable monthly regional emissions inventory, like GEAA-AEIv3.0M, are expected to result in a remarkable improvement in the chemical prediction achieved by air quality models, such as WRF-Chem. This consideration is important, especially in countries where air quality monitoring networks are scarce, and long-term governmental environmental programs are discontinued due to the recurrent economic crisis.

**Supplement**

The supplement related to this article compiles Figure S1 to Figure S17, which show monthly and annual variations for the different subsectors analysed, and two tables (i.e., Table S1 and Table S2), which are available online.

**Data availability**

The GEAA-AEIv3.0M inventory contains spatially distributed monthly emissions for $CO_{2eq}$, $CO_2$, $CH_4$, $N_2O$, CO, NOx, NMVOC, $NH_3$, $SO_2$, $PM_{10}$, $PM_{2.5}$, TSP and BC between 1995 and 2020, and includes the following subsectors: energy production, fugitive emissions from oil and gas production, industrial fuel consumption and production, transport (road, maritime and air), agriculture, livestock production, residential, commercial and biomass burning. The inventory is available as NetCDF files with a spatial resolution of 2.5 km × 2.5 km resolution, between 53° to 73° west longitude and between 21° to 55° south latitude. The files can be openly accessed through the Mendeley Datasets repository at http://dx.doi.org/10.17632/d6xrhpmzdp.1 under a CC-BY 4 license (Puliafito et al., 2021). The main page of the repository



has detailed information on the files hosted, as well as a readme.txt file with specific information to access and interpret the whole dataset.

## Author contributions

S. Enrique Puliafito: Conceptualization, methodology, investigation, formal analysis, original writing & supervising; Tomás R. Bolaño-Ortiz, investigation, visualization, data organization, editing & writing; , Rafael P. Fernandez: formal analysis, supervising & writing; Lucas L. Berná: data organization & investigation, Romina M. Pascual-Flores data organization & investigation, Josefina Urquiza: data organization & investigation, Ana I. López-Noreña: data organization & investigation María F. Tames: editing & writing,

## Competing interests

The authors declare that they have no conflict of interest.

## Acknowledgements

The authors would like to thank Universidad Tecnológica Nacional (UTN) (National Technological University) and Consejo Nacional de Investigaciones Científicas y Técnicas (CONICET) (National Council for Scientific and Technical Investigations) for supporting research activities. This work is granted by UTN IFI Projects PID 1799, 1487 and 4920, CONICET (PIP 112 201101 00673) and Agencia FONCYT (PICT 2016-1115 and 2016-0714). All data requests should be addressed to the first and corresponding author.

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



**Earth System** Discussions
**Science**
**Data**
**FIGURES**

a)
b)

c)
d)

**Figure 1. Spatial coverage and scales used in this inventory: a) Geographical location of Argentina in South America (provinces in**
**white outline); b) main roads; c) districts (black outline) and censal fractions (grey outline); d) spatial gird with districts in**
**background.**

a)

b)

c)

d)

**Figure 2: Location of point sources: a) thermal power plants (districts in white outline); b) manufacturing industries (provinces in white outline). Manuf. code. 2A: cement, calcium, glass, mining; 2B: chemical; 2C: steel, iron, aluminium; 2D: car-painting; 2E. other non-specified, 2H: paper, food, beverages (see Table A3 App.); c) District distribution of annual gasoline sales for year 2019; d) Location of refuelling gas stations and their individual yearly gasoline sales.**


a)

b)

c)

d)

e)

f)

**Figure 3. a)** Evolution of monthly energy consumption by thermal power plants; **b)** GHG emissions evolution (in terms of CO2eq-Gg) from energy consumption at thermal power plants; **c)** Monthly fuel consumption for residential & commercial sector; **d)** Seasonal average fuel consumption for residential & commercial sector for the period 1995-2019. **e)** Annual NOx emissions (in t) from manufacturing activities: **f)** Annual PM10 emissions (in t) for manufacturing activities. Ref. manuf. codes: 2A: cement, calcium, glass, mining; 2B: chemical; 2C: steel, iron, aluminium; 2D: car-painting; 2H: paper, food, beverages (see Table A3, App.).



**Figure 4. a)** Annual animal production for three brides: Bovine, dairy and poultry; **b)** Annual evolution of GHG (in Gg) from for three brides: Bovine (beef production), Bovine (dairy production), other livestock's brides; **c)** Annual evolution of main agriculture indices: Crop production (Gg), cultivated area (kHa) and Use of fertilizers (Gg); **d)** annual emissions of $N_2O$, $NH_3$ and $PM_{10}$ from fertilizers uses and arable lands; **e)** average stational burned area in kHa for the period 1995-2020, according to land type; **f)** annual emissions evolution of $PM_{2.5}$ (kt) for the period 1995-2020, according to land type.

**a)**

**b)**

**c)**

**d)**

**Figure 5. GHG participation by activity for Argentina years 1995 (a) and 2019 (c), (see Table 3); and sectoral SLCP pollutant contribution share of emissions for Argentina: b) year 1995, b) year 2019.** Reference codes are provided in Table 1a.



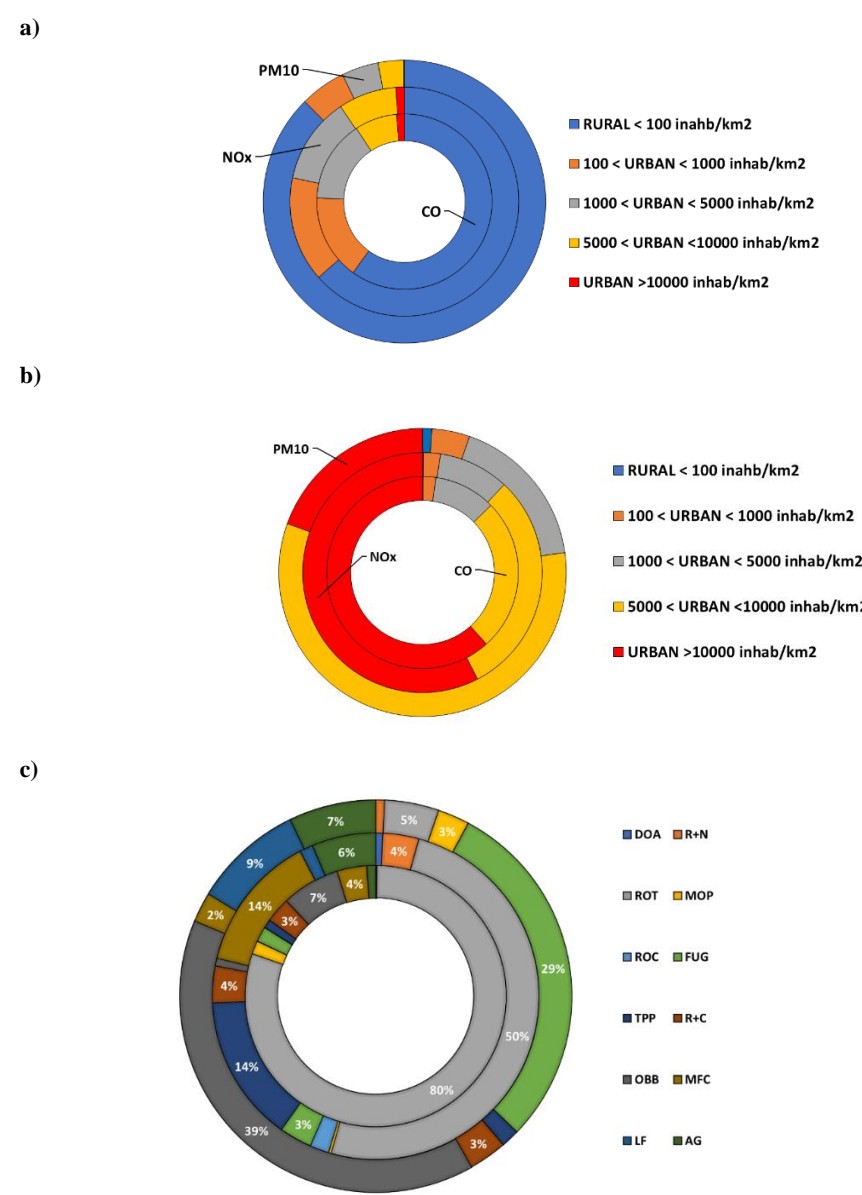


**Figure 6: Annual PM10 (outer ring), NOx (middle ring) and CO (inner ring) emissions distribution according to different classifications: a) total emissions with respect to population density; b) emissions density (kg/km²/year) with respect to urban density, c) total sectoral contribution (see Table 4). Reference codes are provided in Table 1a.**




a)

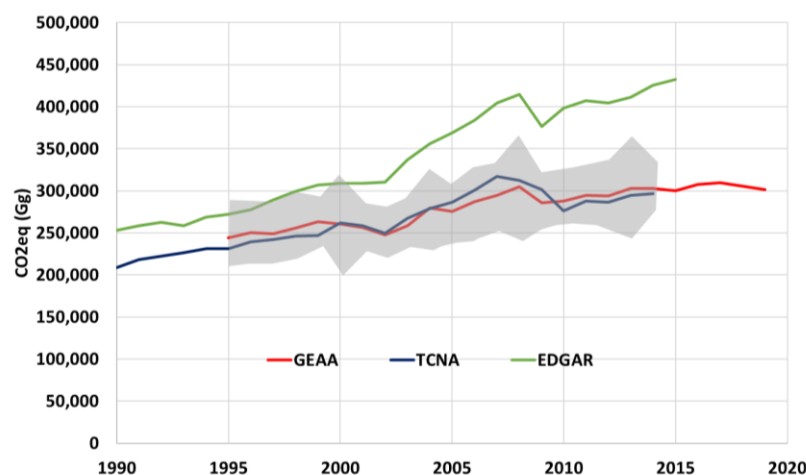

b)

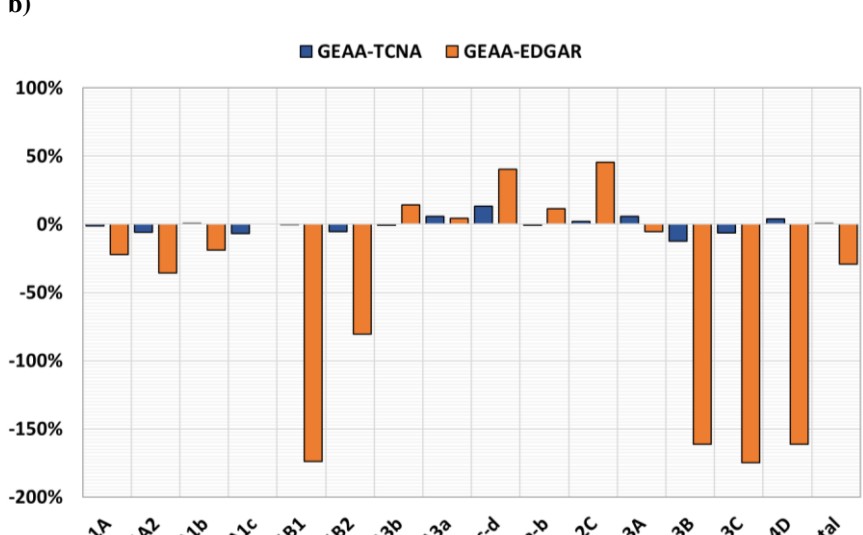

**Figure 7: a) Evolution of total annual CO₂eq-Gg emissions for GEAA (red), TCNA (blue) and EDGAR (green) inventories for Argentina years 1990-2019. (Table 5 and Tables A5 App.); b) Percentage difference in GHG emissions [(GEAA – TCNA)/GEAA and (GEAA – EDGAR)/GEAA] for years 1995 through 2014, for the considered activities. (see also Tables A6 and A7 App.).**

**a)**

**c)**

**b)**

**d)**

**Figure 8: GEAA and EDGAR annual PM$_{10}$ emissions from road transport sector: a) differences (t/year/cell); c) annual series. GEAA and EDGAR annual PM10 emissions from residential + commercial activities: b) differences (t/year/cell); d) annual series. Maps are represented at 0.1×0.1 resolution for year 2015.**


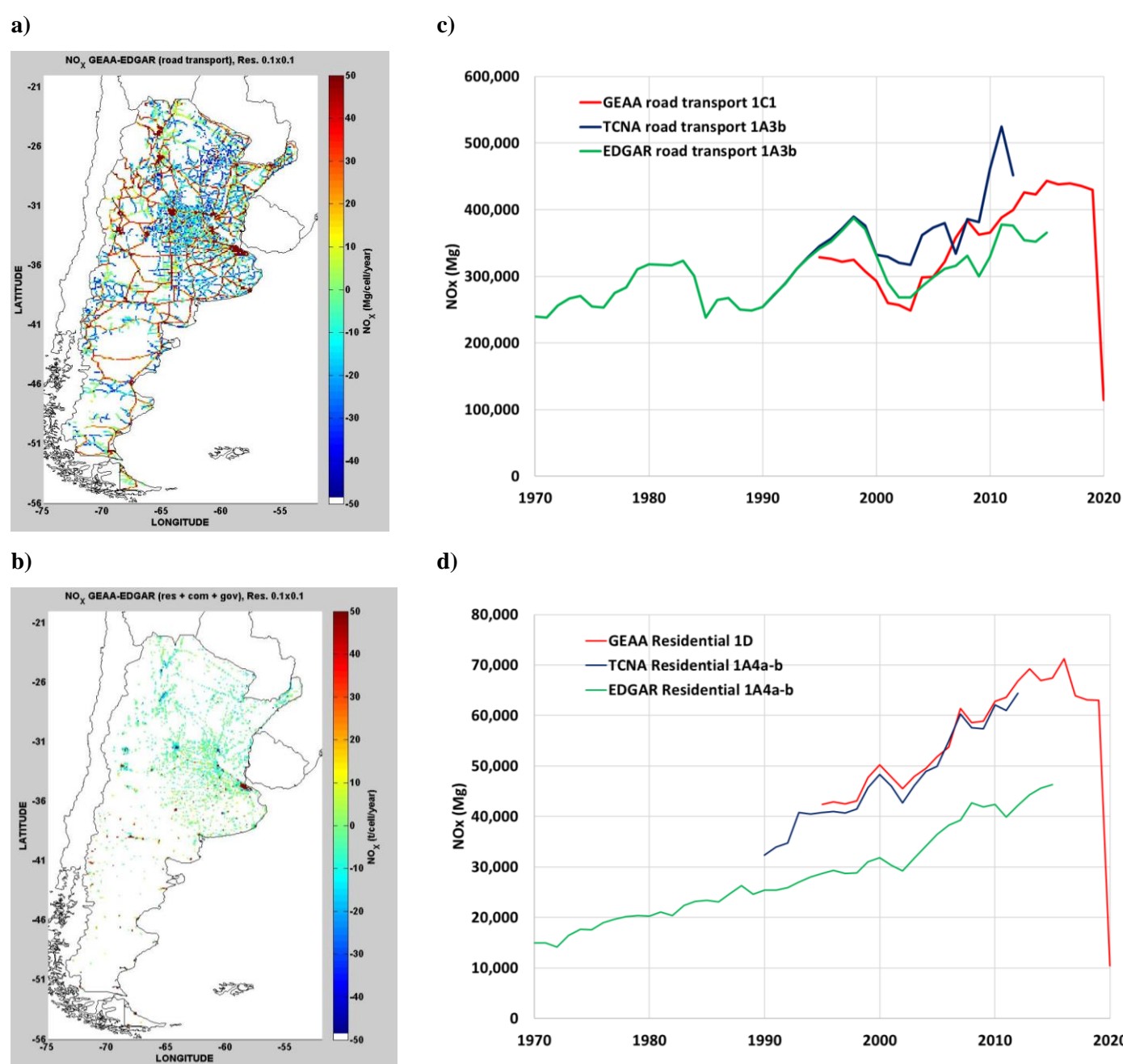

**Figure 9: GEAA and EDGAR annual NOx emissions from road transport sector: a) differences (GEAA-EDGAR) in Mg/year/cell); c) annual series. GEAA and EDGAR annual NOx emissions from residential + commercial activities: b) differences (Mg/year/cell); d) annual series. Maps are represented at 0.1×0.1 resolution for year 2015.**





**Tables**

**Table 1a. Sectors, activities, classification codes and resolution considered in GEAA-AEIv3.0M inventory**

| Sector and Activities | Acronym | IPCC code | Text Section | Period / Resolution | Spatial coverage /resolution |
|---|---|---|---|---|---|
| **Fuel Combustion** | | | | 1995-2020 Monthly | National 0.025° x 0.025° |
| Power and heat production | TPP | 1A1a | 2.3.1 / 3.1 | | |
| Manufacture own energy production | MFC | 1A1b | 2.3.5 / 3.4 | | |
| Fuel Production and transformation | | 1A1b | 2.3.2 / 3.2 | | |
| refinery own consumption, | ROC | 1A1c | 2.3.2 / 3.2 | | |
| oil and gas extraction at wells, | FPR | 1Ab2 | 2.3.2 / 3.2 | | |
| fugitive emissions, venting, and flaring | FUG | | | | |
| Road transportation | ROT | 1A3b | 2.3.3 / 3.3 | | |
| Domestic aviation | DOA | 1A3a | 2.3.3 / 3.3 | | |
| Railroad and navigation | R+N | 1A3c, d | 2.3.3 / 3.3 | | |
| Residential Commercial and Public offices | R+C | 1A4a, b | 2.3.4 / 3.4 | | |
| combustion | | 1A4c | | | |
| Fuel use in Agriculture | FAG | | | | |
| **Manufacture Processes (non-combustion)** | | | | 1995-2020 Monthly | National 0.025° x 0.025° |
| Production of minerals, chemicals, and metals, | MOP | 2B + 2C | 2.3.5 / 3.5 | | |
| pulp/paper/food/drink | | | | | |
| **Agriculture and livestock feeding** | | | | 1995-2020 Yearly | National 0.025° x 0.025° |
| Enteric fermentation, Manure management and | LF | 4A + 4B | 2.3.6 / 3.6 | | |
| Feeding and manure deposited on pasture | | | | | |
| Rice cultivation, Fertilizer application and Crop | AG | 4C + 3C3 | 2.3.6 / 3.6 | | |
| residues | | | | | |
| **Fires** | | | | 1995-2020 Monthly | National 0.025° x 0.025° |
| Biomass & Savanna burning and Fires from LULC | OBB | | 2.3.7 / 3.7 | | |
| Agricultural waste burning | AWB | 4F | 2.3.6 / 3.6 | | |



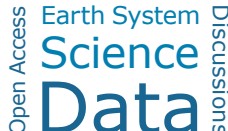



**Table 2b. Sectors, activities, and pollutants considered in GEAA-AEIv3.0M inventory.**

| Sector and Activities | $CO_2$ | $CH_4$ | $N_2O$ | CO | $NH_3$ | $NO_x$ | $SO_2$ | NMVCOC | TSP | $PM_{10}$ | $PM_{2.5}$ | BC |
|---|---|---|---|---|---|---|---|---|---|---|---|---|
| **Fuel Combustion:** | | | | | | | | | | | | |
| Power and heat production | x | x | x | x | | x | x | x | x | x | x | x |
| Fuel Production (incl. fugitive emissions, venting, and flaring) | x | x | x | x | x | x | x | x | x | x | x | x |
| Road transportation | x | x | x | x | x | x | x | x | x | x | x | x |
| Domestic aviation | x | x | x | x | | x | x | x | x | x | x | x |
| Railroad and navigation | x | x | x | x | x | x | x | x | x | x | x | x |
| Residential Commercial and Public offices combustion | x | x | x | x | x | x | x | x | x | x | x | x |
| Fuel use in agriculture | x | x | x | x | | x | x | x | x | x | x | x |
| **Industrial Processes (non-combustion):** | | | | | | | | | | | | |
| Production of minerals, chemicals, and metals, pulp/paper/food/drink | x | x | x | x | x | x | x | x | x | x | x | x |
| **Agriculture and livestock feeding:** | | | | | | | | | | | | |
| Enteric fermentation, Manure management and Feeding and manure deposited on pasture | x | x | x | x | x | x | x | x | x | x | x | x |
| Rice cultivation, Fertilizer application and Crop residues | x | x | x | x | x | x | x | x | x | x | x | x |
| **Fires:** | | | | | | | | | | | | |
| Biomass & Savanna burning and Fires from LULC | x | x | x | x | x | x | x | x | x | x | x | x |
| Agricultural waste burning | x | x | x | x | x | x | x | x | x | x | x | x |



**Table 3. Acronyms used in this text**

| Acronym [&] | Definition | Web page / observation |
|---|---|---|
| National Agencies in Argentina | | |
| ANAC | Argentine Aviation Regulation Agency | https://www.argentina.gob.ar/anac |
| CAMMESA | National Electricity Administration Agency | https://portalweb.cammesa.com/default.aspx |
| CNRT | National Transport Regulation Agency | https://www.argentina.gob.ar/transporte/cnrt |
| MINEM | Energy data base at the Ministry for Energy | https://www.argentina.gob.ar/economia/energia/datos-y-estadisticas |
| INDEC | Statistics and Census Bureau | https://www.indec.gob.ar/ |
| MAyGN | Ministry of Agriculture and Livestock | https://www.magyp.gob.ar/datosabiertos/ |
| SENASA | Animal Health Control Agency | https://www.argentina.gob.ar/senasa |
| INTA | National Agricultural Research Institute | https://www.argentina.gob.ar/inta |
| Models, software, inventories, and international databases | | |
| GHG | Greenhouse Gases | |
| GWP100 | Global Potential Warming for 100 years | CH4: 28; N2O: 298 |
| IPCC | Intergovernmental Panel on Climate Change | https://archive.ipcc.ch/index.htm |
| AQM | Air quality models | |
| AEI | Atmospheric Emissions Inventory | |
| GEAA-AEIV2.5 | High Resolution Atmospheric Emissions Inventory for Argentina | This study and Puliafito et al, (2015,2017,2020) |
| TCNA | Third Argentine National Greenhouse Gases Inventory | https://www.argentina.gob.ar/ambiente/cambio-climatico/tercer-informe-bienal |
| WRF-Chem | Weather Research and Forecasting with Chemistry | Grell et al, (2005) https://www2.acom.ucar.edu/wrf-chem |
| EDGAR-HTAP | Emissions Database for Global Atmospheric Research – Hemispheric Transport of Air Pollution | https://edgar.jrc.ec.europa.eu/htap.php |
| EMEP | European Monitoring and Evaluation Program Guidebook | https://www.eea.europa.eu/publications/emep-eea-guidebook-2019 |
| EMEP | European Monitoring and Evaluation Program Guidebook | https://www.eea.europa.eu/publications/emep-eea-guidebook-2019 |
| MODIS | MODerate resolution Imaging Spectroradiometer | https://modis.gsfc.nasa.gov/ |
| GIS | Geographical Information System | Software for spatial database handling |

**[&] additional acronyms are compiled in Table A2, Appen.**





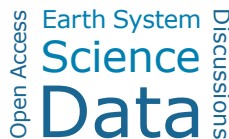

**Table 4. Summary of annual emissions for years 2019 and 1995 for Argentina**

| ACTIVITY | $CO_2$ (Gg) | $CH_4$ (Mg) | $N_2O$ (Mg) | $NO_x$ (Mg) | CO (Mg) | NMVOC (Mg) | $SO_2$ (Mg) | $NH_3$ (Mg) | TSP (Mg) | $PM_{10}$ (Mg) | $PM_{2.5}$ (Mg) | BC (Mg) |
|---|---|---|---|---|---|---|---|---|---|---|---|---|
| TPP 2019 | 35,652.80 | 5,143.36 | 1,448.05 | 124,549.38 | 43,701.34 | 6,730.23 | 10,475.70 | 106.24 | 942.12 | 848.98 | 686.22 | 69.86 |
| TPP 1995 | 17,791.80 | 2,191.66 | 617.17 | 58,083.69 | 18,799.47 | 2,913.75 | 29,216.58 | 89.04 | 1,070.07 | 833.42 | 571.91 | 111.59 |
| MFC 2019 | 24,992.33 | 2,208.53 | 295.41 | 41,752.94 | 18,022.92 | 2,692.70 | 2,733.86 | - | 10,133.40 | 9,588.97 | 9,045.70 | 2,447.42 |
| MFC 1995 | 26,250.01 | 2,824.24 | 396.77 | 41,677.14 | 17,262.56 | 3,179.19 | 5,514.90 | - | 14,358.78 | 13,417.68 | 12,263.79 | 3,347.67 |
| ROC 2019 | 6,098.84 | 126.61 | 13.99 | 16,262.00 | 2,579.28 | 583.71 | 488.84 | - | 162.56 | 124.63 | 87.83 | 13.89 |
| ROC 1995 | 3,771.45 | 96.22 | 9.62 | 9,846.93 | 1,194.96 | 297.65 | 601.77 | - | 122.81 | 95.36 | 51.33 | 7.83 |
| FPR 2019 | 9,930.03 | 191.13 | 20.53 | 27,488.27 | 70,852.34 | 112,107.44 | 2,904.95 | - | 472.37 | 472.37 | 472.37 | 109.01 |
| FPR 1995 | 4,823.77 | 88.04 | 9.01 | 13,034.35 | 34,401.38 | 46,431.36 | 423.20 | - | 225.33 | 225.33 | 225.33 | 52.00 |
| FUG 2019 | 3,677.26 | 483,400.36 | 70.61 | 369.51 | 369.59 | 167.81 | 0.19 | 0.30 | 9.90 | 9.93 | 9.90 | 2.37 |
| FUG 1995 | 2,854.72 | 232,373.71 | 30.47 | 291.38 | 291.45 | 149.07 | 0.15 | 0.23 | 7.81 | 7.83 | 7.81 | 1.87 |
| ROT 2019 | 49,113.18 | 15,479.65 | 3,665.54 | 429,428.25 | 3,115,081.40 | 405,326.79 | 13,305.08 | 13,632.44 | 14,670.69 | 11,736.55 | 10,562.89 | 2,676.82 |
| ROT 1995 | 40,256.52 | 9,451.45 | 2,908.92 | 328,342.02 | 2,369,115.10 | 307,051.85 | 12,371.08 | 10,519.94 | 13,814.63 | 11,051.70 | 9,946.53 | 4,974.70 |
| DOA 2019 | 1,778.89 | 12.44 | 49.76 | 6,219.91 | 2,487.96 | 1,243.98 | 22.14 | 1,128.29 | 19.90 | 12.44 | 0.56 | 0.72 |
| DOA 1995 | 1,547.02 | 10.82 | 43.27 | 5,409.16 | 2,163.66 | 1,081.83 | 19.26 | 981.22 | 17.31 | 10.82 | 0.49 | 0.56 |
| R+N 2019 | 1,307.21 | 119.02 | 33.30 | 30,815.96 | 3,350.27 | 1,283.33 | 7,345.94 | 3.04 | 2,198.82 | 2,192.28 | 1,985.79 | 1,290.77 |
| R+N 1995 | 411.51 | 38.61 | 10.48 | 8,674.78 | 1,180.56 | 478.38 | 1,854.75 | 0.96 | 513.66 | 508.55 | 463.77 | 301.45 |
| R+C 2019 | 29,093.12 | 3,109.03 | 120.33 | 30,828.86 | 121,365.88 | 9,457.17 | 1,964.15 | 126.51 | 9,198.03 | 8,623.35 | 8,393.97 | 654.35 |
| R+C 1995 | 21,148.13 | 2,540.13 | 126.00 | 22,271.06 | 135,381.73 | 12,586.84 | 4,158.82 | 78.89 | 14,247.03 | 13,638.59 | 13,093.28 | 1,173.86 |
| FAG 2019 | 11,161.22 | 446.45 | 253.58 | 48,885.36 | 41,242.62 | 6,903.12 | 612.70 | 133.84 | 5,987.40 | 5,353.44 | 5,353.44 | 2,465.40 |
| FAG 1995 | 8,519.00 | 340.76 | 193.55 | 37,312.65 | 31,479.19 | 5,268.93 | 467.66 | 102.15 | 4,569.99 | 4,086.11 | 4,086.11 | 1,881.76 |
| MOP 2019 | 13,150.74 | 116.27 | 562.25 | 2,399.50 | 62,061.61 | 25,880.67 | 6,256.51 | 4,015.92 | 22,565.27 | 8,155.49 | 4,580.94 | 90.16 |
| MOP 1995 | 8,506.78 | 257.58 | 463.68 | 1,996.66 | 29,614.58 | 20,754.96 | 7,600.02 | 976.95 | 9,865.09 | 3,361.22 | 1,987.79 | 69.96 |
| LF 2019 | - | 2,781,099.06 | 87,068.47 | 6,673.93 | - | 204,075.53 | - | 211,634.17 | 89,789.65 | 27,806.44 | 11,327.39 | - |
| LF 1995 | - | 3,084,156.16 | 81,859.67 | 5,530.74 | - | 209,738.66 | - | 154,913.12 | 50,213.95 | 19,964.21 | 11,541.43 | - |
| AG 2019 | 645.99 | 39,646.34 | 21,760.71 | 67,825.59 | - | 15,169.70 | - | 529,442.66 | 3,792.42 | 3,033.94 | 2,275.45 | - |
| AG 1995 | 83.24 | 26,291.25 | 3,205.65 | 9,991.65 | - | 7,930.75 | - | 86,602.95 | 1,982.69 | 1,586.15 | 1,189.61 | - |
| AWB 2019 | 1,551.18 | 3,327.46 | 68.51 | 5,186.92 | 65,961.93 | 3,914.65 | 831.36 | 439.70 | 9,623.73 | 479.19 | 6,161.53 | 7,726.41 |
| AWB 1995 | 1,731.25 | 3,713.72 | 76.46 | 5,789.04 | 73,619.08 | 4,369.08 | 913.36 | 490.74 | 10,558.49 | 524.62 | 6,759.89 | 8,478.19 |
| OBB 2019 | 4,627.87 | 74,128.96 | 5,336.02 | 6,083.67 | 214,146.38 | 2,679.91 | 874.61 | 2,631.45 | 48,672.18 | 20,918.91 | 10,720.87 | 1,440.13 |
| OBB 1995 | 8,079.03 | 128,727.05 | 9,185.10 | 10,059.21 | 372,831.42 | 4,602.62 | 1,476.97 | 4,610.87 | 84,698.39 | 35,006.35 | 18,020.96 | 2,411.27 |
| TOT. 2019 | 192,780.67 | 3,408,554.67 | 120,767.07 | 844,770.06 | 3,761,223.52 | 798,216.75 | 47,816.05 | 763,294.56 | 218,238.45 | 99,356.93 | 71,664.86 | 18,987.29 |
| TOT. 1995 | 145,774.24 | 3,493,101.41 | 99,135.85 | 558,310.45 | 3,087,335.13 | 626,834.95 | 64,618.50 | 259,367.06 | 206,266.03 | 104,317.95 | 80,210.04 | 22,812.71 |

Ref (see Table 1a): TPP: Power Plants, MFC: Manufacturing own fuel consumption, ROC: Refinery own consumption, FPR: Fuel production, FUG: Fugitive, venting and flare, ROT: Road transport, DOA: Domestic Aviation, R+N: Railroad and navigation, R+C: Residential and commercial, FAG: Fuel use in agriculture, MOP: Manufacturing own process, LF: Livestock feeding, AG: Agriculture, AWB: Agriculture waste burning, OBB: Open biomass burning.




**Table 5. Emission distribution of CO and NOx according to population density for year 2019.**

| Density (d) inhab /km² | RURAL d < 100 | URBAN VERY LOW 100 < d < 1000 | URBAN LOW 1000 < d < 5000 | URBAN MEDIUM 5000 < d <10000 | URBAN HIGH d >10000 | TOTAL |
|---|---|---|---|---|---|---|
| N.of cells | 445,917.00 | 6,508.00 | 1,285.00 | 269.00 | 21.00 | 454,000.00 |
| AREA (km2) | 2,786,981.25 | 40,675.00 | 8,031.25 | 1,681.25 | 131.25 | 2,837,500.00 |
| Popul. 2020 | 659,690 | 11,010,333 | 18,590,350 | 12,658,283 | 2,115,971 | 45,034,627 |
| % Pop. | 1.5% | 24.4% | 41.3% | 28.1% | 4.7% | 100.0% |
| CO (Mg/year) | | | | | | |
| DOA | 939.49 | 382.50 | 362.07 | 803.90 | - | 2,487.96 |
| R+N | 1,228.38 | 763.46 | 773.51 | 503.44 | 81.48 | 3,350.27 |
| ROT | 1,891,902.61 | 504,092.42 | 418,483.32 | 248,576.30 | 52,026.75 | 3,115,081.40 |
| MOP | 4,037.09 | 2,366.40 | 55,431.16 | 226.96 | - | 62,061.61 |
| ROC | 194.27 | 288.94 | 1,759.29 | 336.78 | - | 2,579.28 |
| FUG | 2,640.05 | 13,606.92 | 42,161.21 | 12,813.76 | - | 71,221.94 |
| TPP | 23,039.50 | 10,272.25 | 3,020.20 | 7,369.39 | - | 43,701.34 |
| R+C | 3,139.98 | 51,526.33 | 41,609.18 | 20,600.59 | 4,489.79 | 121,365.88 |
| OBB | 271,743.78 | 5,518.57 | 2,295.39 | 550.57 | - | 280,108.31 |
| MFC | 91,669.96 | 21,091.52 | 20,231.07 | 11,176.89 | 69.92 | 144,239.36 |
| AG | 27,000.48 | 14,242.14 | - | - | - | 41,242.62 |
| total | 2,317,535.60 | 624,151.46 | 586,126.39 | 302,958.57 | 56,667.94 | 3,887,439.96 |
| NOX (Mg/year) | | | | | | |
| DOA | 3,062.42 | 638.05 | 703.51 | 1,815.92 | - | 6,219.91 |
| R+N | 12,571.81 | 7,438.30 | 6,515.27 | 3,707.83 | 582.76 | 30,815.96 |
| ROT | 271,442.41 | 69,834.64 | 52,841.31 | 29,604.42 | 5,705.48 | 429,428.25 |
| MOP | 855.57 | 536.36 | 948.55 | 59.02 | - | 2,399.50 |
| ROC | 1,220.47 | 1,802.55 | 11,090.60 | 2,148.39 | - | 16,262.00 |
| FUG | 25,576.21 | 1,458.16 | 631.49 | 191.92 | - | 27,857.78 |
| TPP | 65,471.13 | 31,357.67 | 7,744.36 | 19,976.22 | - | 124,549.38 |
| R+C | 125.16 | 4,279.10 | 13,112.24 | 9,268.83 | 4,043.54 | 30,828.86 |
| LF | 5,943.46 | 631.22 | 92.73 | 6.52 | - | 6,673.93 |
| AG | 72,850.49 | 43,860.46 | - | - | - | 116,710.95 |
| OBB | 10,882.44 | 254.92 | 106.81 | 26.41 | - | 11,270.58 |
| MFC | 30,592.88 | 8,978.06 | 11,073.35 | 3,480.78 | 32.79 | 54,157.85 |
| total | 500,594.44 | 171,069.48 | 104,860.21 | 70,286.27 | 10,364.57 | 857,174.97 |

The total Argentine population, surface extension, total emission and emission density are classified according to the mean urban density within each cell.

Ref (see Table 1a): PP: Power Plants, MFC: Manufacturing own fuel consumption, ROC: Refinery own consumption, FPR: Fuel production, FUG: Fugitive, venting and flare, ROT: Road transport, DOA. Domestic Aviation, R+N: Railroad and navigation, R+C (NG): Residential and commercial (natural gas), R+C (OF) Residential and commercial (other fuels), FAG: Fuel use in agriculture, MOP: Manufacturing own 1065 process, LF: Livestock feeding, AG: Agriculture, AWB: Agriculture waste burning, OBB. Open biomass burning.





**Table 6. GEAA-AEIv3.0M inventory: annual GHG emissions values (CO2eq-Gg) for Argentina**

| GEAA | TPP | MFC | ROC | FPR | FUG | ROT | DOA | R+N | R+C | MOP | LF | AG | AWB | OBB |
|---|---|---|---|---|---|---|---|---|---|---|---|---|---|---|
| 1,995 | 18,030.51 | 18,024.89 | 3,776.73 | 4,828.66 | 8,673.14 | 41,359.67 | 1,560.18 | 415.60 | 29,834.39 | 8,651.39 | 101,498.09 | 1,695.80 | 115.63 | 5,955.34 |
| 1,996 | 21,090.39 | 18,493.91 | 3,899.74 | 5,226.66 | 9,046.74 | 41,807.13 | 1,367.99 | 441.06 | 30,641.33 | 10,205.50 | 99,446.99 | 2,573.35 | 112.54 | 5,929.80 |
| 1,997 | 19,530.46 | 20,579.40 | 5,122.42 | 5,707.75 | 10,135.26 | 42,543.93 | 1,298.71 | 455.59 | 30,136.13 | 9,779.01 | 94,889.85 | 2,605.04 | 124.31 | 6,300.14 |
| 1,998 | 20,970.76 | 21,245.05 | 5,506.26 | 6,580.33 | 11,136.94 | 43,878.61 | 1,501.11 | 488.51 | 30,178.06 | 10,676.77 | 95,701.14 | 2,463.19 | 137.69 | 5,441.60 |
| 1,999 | 25,468.28 | 19,497.20 | 4,358.56 | 6,380.70 | 10,991.23 | 42,084.17 | 1,668.87 | 467.98 | 32,432.13 | 10,760.03 | 100,296.06 | 3,446.39 | 131.03 | 5,424.73 |
| 2,000 | 25,181.91 | 19,078.08 | 3,853.25 | 6,693.93 | 10,969.48 | 40,900.27 | 1,496.43 | 461.73 | 33,340.89 | 11,828.29 | 97,815.89 | 3,502.92 | 121.34 | 5,419.34 |
| 2,001 | 18,877.56 | 18,293.31 | 3,969.18 | 6,950.57 | 11,619.47 | 36,373.21 | 1,232.84 | 613.28 | 31,450.41 | 11,691.38 | 99,192.43 | 3,508.62 | 122.61 | 12,628.66 |
| 2,002 | 15,883.52 | 19,958.15 | 4,665.92 | 7,165.76 | 11,764.03 | 35,117.90 | 1,061.34 | 425.57 | 29,312.27 | 12,555.16 | 97,176.91 | 3,553.81 | 117.99 | 8,805.61 |
| 2,003 | 18,080.59 | 20,919.28 | 5,248.45 | 7,481.12 | 12,976.87 | 35,351.22 | 1,002.70 | 551.27 | 32,324.68 | 12,147.16 | 97,348.52 | 4,306.07 | 133.46 | 10,690.27 |
| 2,004 | 24,281.84 | 21,736.17 | 4,448.29 | 7,389.46 | 12,586.92 | 40,308.29 | 1,140.46 | 753.33 | 35,537.13 | 13,746.51 | 105,015.93 | 5,012.78 | 128.42 | 7,611.06 |
| 2,005 | 27,169.90 | 21,790.57 | 4,130.15 | 7,553.08 | 11,885.09 | 38,917.27 | 1,165.37 | 792.22 | 36,624.12 | 14,514.90 | 100,023.34 | 4,650.46 | 134.97 | 6,443.59 |
| 2,006 | 28,942.39 | 23,209.38 | 4,328.69 | 7,823.74 | 11,986.98 | 41,986.89 | 1,061.69 | 895.38 | 37,901.64 | 15,845.14 | 100,422.99 | 5,471.65 | 146.62 | 7,113.35 |
| 2,007 | 32,958.02 | 23,541.34 | 4,513.82 | 8,606.33 | 12,166.84 | 46,734.03 | 1,123.93 | 544.59 | 38,920.64 | 15,917.05 | 97,298.78 | 6,581.49 | 143.39 | 5,779.05 |
| 2,008 | 37,300.44 | 23,525.38 | 4,437.66 | 9,257.86 | 12,058.61 | 49,840.10 | 1,239.22 | 540.22 | 38,358.94 | 15,464.60 | 98,494.27 | 6,351.81 | 142.49 | 9,526.25 |
| 2,009 | 34,346.91 | 21,457.59 | 5,138.60 | 9,764.13 | 11,716.86 | 46,221.94 | 1,277.76 | 513.63 | 40,947.08 | 13,012.09 | 89,388.62 | 4,777.49 | 133.41 | 7,002.68 |
| 2,010 | 36,117.65 | 22,824.46 | 6,033.41 | 9,309.83 | 11,428.79 | 46,971.09 | 1,413.33 | 1,669.57 | 40,637.21 | 15,014.81 | 84,718.60 | 6,615.52 | 116.99 | 5,084.54 |
| 2,011 | 40,519.30 | 23,583.76 | 5,728.80 | 9,267.38 | 11,028.60 | 49,243.13 | 1,400.63 | 2,094.96 | 38,802.34 | 16,199.11 | 85,809.80 | 6,113.89 | 117.58 | 4,842.19 |
| 2,012 | 43,707.73 | 19,629.01 | 6,108.69 | 9,349.48 | 10,902.32 | 49,563.61 | 1,400.28 | 1,970.54 | 40,325.49 | 15,791.57 | 83,684.63 | 6,351.81 | 112.44 | 5,070.32 |
| 2,013 | 42,962.41 | 20,094.33 | 6,075.64 | 9,322.25 | 10,833.52 | 52,138.47 | 1,427.47 | 1,590.85 | 44,862.77 | 16,415.74 | 84,983.24 | 5,561.22 | 95.37 | 6,411.52 |
| 2,014 | 43,761.61 | 17,277.11 | 6,022.37 | 9,668.87 | 10,506.61 | 51,271.14 | 1,479.39 | 1,591.88 | 42,975.00 | 16,749.46 | 92,146.92 | 5,890.00 | 100.09 | 3,619.45 |
| 2,015 | 45,928.68 | 17,204.82 | 6,126.87 | 10,191.02 | 11,000.56 | 53,703.80 | 1,410.44 | 1,633.40 | 43,501.65 | 15,050.95 | 86,008.21 | 4,600.55 | 88.66 | 3,899.49 |
| 2,016 | 46,640.23 | 15,175.80 | 6,212.59 | 10,696.76 | 11,273.33 | 52,378.63 | 1,505.14 | 1,636.37 | 45,168.89 | 14,895.51 | 92,918.38 | 6,886.05 | 90.64 | 2,034.04 |
| 2,017 | 42,936.05 | 16,557.00 | 6,220.11 | 10,256.56 | 11,104.91 | 51,637.09 | 1,589.45 | 1,405.43 | 42,385.01 | 16,071.67 | 94,683.67 | 6,784.85 | 103.72 | 8,128.27 |
| 2,018 | 41,319.14 | 17,483.46 | 6,723.78 | 10,044.87 | 11,492.52 | 51,033.57 | 1,704.82 | 1,496.32 | 40,351.06 | 16,332.22 | 94,930.79 | 7,582.61 | 107.13 | 5,297.25 |
| 2,019 | 36,212.91 | 18,950.36 | 6,106.18 | 9,940.93 | 15,783.31 | 50,592.50 | 1,794.03 | 1,320.11 | 40,454.66 | 13,321.19 | 95,473.88 | 8,121.84 | 103.60 | 3,443.36 |
| 2,020 | 12,472.95 | 6,967.85 | 4,016.53 | 8,212.67 | 10,922.97 | 13,819.76 | 291.27 | 284.23 | 8,719.71 | 325.96 | | | | 1,304.43 |

Ref: (Table 1a) TPP: Power Plants, MFC: Manufacturing own fuel consumption, ROC: Refinery own consumption, FPR: Fuel production, FUG: Fugitive, venting and flare, ROT: Road transport, DOA. Domestic Aviation, R+N: Railroad and navigation, R+C Residential and commercial, MOP: Manufacturing own process, LF: Livestock feeding, AG: Agriculture, AWB: Agriculture waste burning, OBB. Open biomass burning



# APPENDIX

## TABLES

**Table A1. Argentine inventories developed at the Group for Atmospheric and Environmental Studies (GEAA)**

| Name | Sectors | Species | Extension/ Temporal /Resolution | Reference |
|---|---|---|---|---|
| GEAA-AEIv1.0A | Road transport sector | $CO_2$, $CH_4$, CO, NOx, NMVOC, TSP, $PM_{10}$, $PM_{2.5}$ | Argentina, annual 2014, 9 × 9 km | Puliafito et al., (2015) |
| GEAA-AEIv2.0A | Public electricity and heat production, oil refining, fugitive emissions from oil and gas production, domestic aviation, road transport, rail and inland navigation, residential sector, cement production | $CO_2$, $CH_4$, $N_2O$, CO, NOx, NMVOC, TSP, $PM_{10}$, $PM_{2.5}$ | Argentina, annual 2016, 0.025° × 0.025° | Puliafito et al., (2017) |
| GEAA-AEIv3.0A | Public electricity and heat production, oil refining, fugitive emissions from oil and gas production, domestic aviation, road transport, rail and inland navigation, residential sector, cement production, agriculture, livestock production, biomass burning. | $CO_2$, $CH_4$, $N_2O$, CO, NOx, NMVOC, $NH_3$, TSP, $PM_{10}$, $PM_{2.5}$, BC | Argentina, annual, 2016, 0.025° × 0.025° | Puliafito et al., (2020a, 2020b) |

**Table A2. Other acronyms used in this text**

| Acronym | Definition | Web page / observation |
|---|---|---|
| | Fuels and technology considered in power plants | |
| CC | Combined cycle | Power plant technology |
| TV | Turbo steam | Power plant technology |
| TG | Turbo gas | Power plant technology |
| DI | Diesel Engine | Power plant technology |
| NG | Natural Gas | Fuel |
| FO | Heavy fuel oil | Fuel |
| GO | Gasoil | Fuel |
| CM | Mineral coal, carbon, charcoal | Fuel |
| BD | Biodiesel | Fuel |
| | Transport variables | |
| RGS | Refueling Gas Stations | Loading fuel stations for vehicles |
| VKT | Vehicle kilometer transported (v-km) | Passenger transport index |
| TKT | Ton kilometer transported (t-km) | Freight transport index |
| PKT | Passenger kilometer transported (p-km) | Public transport index |
| LTO | Landing and take-off | Aviation index |
| FO | Heavy fuel oil | Fuel for navigation |
| CNG | Compressed natural Gas | Fuel |
| NA | Gasoline | Fuel |
| GO | Gasoil | Fuel |
| AK | Kerosene for aviation | Jet fuel for aviation |
| AG | Gasoline for aviation | Fuel for aviation |



**Table A3. List of industrial activities**

| Number | Code | Activity | Number | Code | Activity |
|---|---|---|---|---|---|
| 1 | 2.C.1 | steel-iron | 24 | 2.B.10 | pet |
| 2 | 2.C.3 | aluminium | 25 | 2.B.10 | polyethylene high density |
| 3 | 2.B.4 | benzoic acid | 26 | 2.B.10 | polyethylene |
| 4 | 2.B.4 | acetaldehiyde | 27 | 2.B.10 | polypropylene |
| 5 | 2.B.4 | acetic acid | 28 | 2.B.10 | ammonium sulphate |
| 6 | 2.B.4 | ethyl acetate | 29 | 2.B.7 | carbon sulfide |
| 7 | 2.B.4 | acetone | 30 | 2.B.4 | toluene |
| 8 | 2.B.4 | n-butyl acetate | 31 | 2.B.10 | urea |
| 9 | 2.B.2 | nitric acid | 32 | 2.H.1 | paper-bisulfite |
| 10 | 2.B.4 | salicylic acid | 33 | 2.H.1 | paper-kraft |
| 11 | 2.B.4 | alcohol | 34 | 2.H.1 | paper-pulp |
| 12 | 2.B.1 | ammonia | 35 | 2.H.2 | vegetable oil |
| 13 | 2.B.4 | aromatics-btx | 36 | 2.H.2 | food-poultry |
| 14 | 2.D.3 | asphalt | 37 | 2.H.2 | sugar |
| 15 | 2.D.3 | asphalt roof | 38 | 2.H.2 | Beverage |
| 16 | 2.D.3 | asphalt roads | 39 | 2.A.2 | calcium lime |
| 17 | 2.B.10 | sulfuric acid | 40 | 2.A.1 | cement |
| 18 | 2.B.2 | benzene | 41 | 2.D.3 | car painting |
| 19 | 2.B.7 | sodium carbonate | 42 | 2.B.5 | calcium carbide |
| 20 | 2.B.10 | chlorine | 43 | 2.A.3 | glass |
| 21 | 2.B.10 | ethylene | 44 | 2.A.2 | calcium lime |
| 22 | 2.B.10 | nylon | 45 | 2.A.1 | cement |
| 23 | 2.B.10 | other-chemical | | | |






**Table A4 Summary of annual pollutants emissions for Argentina during December 2019 and December 1995**

| ACTIVITY | CO2 (Gg) | CH4 (Mg) | N2O (Mg) | NOx (Mg) | CO (Mg) | NMVOC (Mg) | SO2 (Mg) | NH3 (Mg) | TSP (Mg) | PM10 (Mg) | PM2.5 (Mg) | BC (Mg) |
|---|---|---|---|---|---|---|---|---|---|---|---|---|
| TPP 2019 | 3,123.90 | 449.66 | 127.10 | 10,823.88 | 3,762.27 | 585.25 | 1,359.73 | 7.44 | 74.98 | 68.64 | 51.64 | 3.65 |
| TPP 1995 | 1,685.47 | 226.82 | 63.73 | 5,733.74 | 1,961.89 | 300.09 | 1,406.02 | 6.89 | 71.35 | 58.62 | 43.98 | 7.14 |
| MFC 2019 | 2,493.85 | 225.83 | 30.09 | 4,129.33 | 1,806.67 | 273.80 | 258.12 | - | 1,031.55 | 977.86 | 928.21 | 251.43 |
| MFC 1995 | 2,199.21 | 236.32 | 33.13 | 3,451.02 | 1,442.08 | 266.12 | 448.36 | - | 1,201.29 | 1,122.91 | 1,026.84 | 280.55 |
| ROC 2019 | 525.14 | 10.50 | 1.18 | 1,402.42 | 228.47 | 51.37 | 35.24 | - | 13.48 | 10.30 | 7.63 | 1.20 |
| ROC 1995 | 304.84 | 7.11 | 0.71 | 797.42 | 92.21 | 24.79 | 35.30 | - | 8.96 | 6.90 | 4.32 | 0.61 |
| FPR 2019 | 718.79 | 40.40 | 6.30 | 256.74 | 611.91 | 24,136.47 | 286.57 | - | 0.73 | 0.73 | 0.73 | 0.17 |
| FPR 1995 | 392.39 | 25.31 | 2.06 | 80.30 | 314.88 | 9,092.22 | 45.84 | - | 0.42 | 0.42 | 0.42 | 0.10 |
| FUG 2019 | 234.98 | 38,924.24 | 5.64 | 0.01 | 8.35 | 14.83 | 0.02 | 0.03 | 0.24 | 0.24 | 0.24 | 0.06 |
| FUG 1995 | 233.36 | 18,933.68 | 2.45 | 0.00 | 9.33 | 13.08 | 0.01 | 0.02 | 0.26 | 0.26 | 0.26 | 0.07 |
| ROT 2019 | 4,271.50 | 1,369.27 | 322.68 | 37,707.75 | 177,927.01 | 35,948.09 | 1,138.12 | 1,239.09 | 1,248.63 | 998.90 | 899.01 | 229.11 |
| ROT 1995 | 3,781.77 | 889.39 | 280.08 | 31,279.36 | 147,935.03 | 29,954.89 | 1,142.29 | 1,003.80 | 1,268.36 | 1,014.69 | 913.22 | 456.35 |
| DOA 2019 | 147.70 | 1.03 | 4.13 | 516.43 | 206.57 | 103.29 | 93.68 | 1.84 | 1.65 | 1.03 | 0.05 | 0.15 |
| DOA 1995 | 174.47 | 1.22 | 4.88 | 610.04 | 244.02 | 122.01 | 110.66 | 2.17 | 1.95 | 1.22 | 0.05 | 0.18 |
| R+N 2019 | 77.66 | 7.11 | 1.98 | 203.61 | 1,793.42 | 0.18 | 78.94 | 419.77 | 124.14 | 123.64 | 112.11 | 72.87 |
| R+N 1995 | 34.96 | 3.29 | 0.89 | 101.44 | 727.53 | 0.08 | 41.32 | 153.37 | 42.00 | 41.54 | 37.92 | 24.65 |
| R+C 2019 | 1,100.72 | 117.30 | 4.54 | 1,168.39 | 4,541.12 | 352.27 | 74.51 | 4.79 | 340.82 | 319.28 | 310.95 | 24.10 |
| R+C 1995 | 653.28 | 77.45 | 3.85 | 688.12 | 4,123.98 | 383.28 | 126.58 | 2.45 | 433.65 | 415.14 | 398.55 | 35.73 |
| FAG 2019 | 885.02 | 35.40 | 20.11 | 3,876.33 | 3,270.30 | 547.38 | 48.58 | 10.61 | 474.77 | 424.50 | 424.50 | 195.49 |
| FAG 1995 | 743.69 | 29.75 | 16.90 | 3,257.29 | 2,748.05 | 459.96 | 40.83 | 8.92 | 398.95 | 356.71 | 356.71 | 164.27 |
| MOP 2019 | 1,084.50 | 10.47 | 44.65 | 201.84 | 4,871.35 | 825.26 | 512.38 | 352.43 | 1,621.51 | 648.70 | 371.99 | 7.29 |
| MOP 1995 | 779.38 | 22.11 | 36.55 | 164.31 | 2,534.76 | 711.51 | 647.88 | 81.17 | 814.25 | 310.90 | 184.44 | 6.53 |
| LF 2019 | - | 231,758.26 | 7,255.71 | 556.16 | | 17,636.18 | - | 17,006.29 | 7,482.47 | 2,317.20 | 943.95 | - |
| LF 1995 | - | 257,013.01 | 6,821.64 | 460.90 | | 12,909.43 | - | 17,478.22 | 4,184.50 | 1,663.68 | 961.79 | - |
| AG 2019 | 53.83 | 3,303.86 | 1,813.39 | 5,652.13 | | 1,264.14 | - | 44,120.22 | 316.04 | 252.83 | 189.62 | - |
| AG 1995 | 6.94 | 2,190.94 | 267.14 | 832.64 | | 660.90 | - | 7,216.91 | 165.22 | 132.18 | 99.13 | - |
| AWB 2019 | 129.27 | 277.29 | 5.71 | 432.24 | 5,496.83 | 326.22 | 69.28 | 36.64 | 801.98 | 39.93 | 513.46 | 643.87 |
| AWB 1995 | 144.27 | 309.48 | 6.37 | 482.42 | 6,134.92 | 364.09 | 76.11 | 40.90 | 879.87 | 43.72 | 563.32 | 706.52 |
| OBB 2019 | 366.66 | 1,237.97 | 20.40 | 574.33 | 22,179.19 | 274.71 | 71.02 | 332.25 | 6,429.18 | 4,094.06 | 2,003.57 | 144.56 |
| OBB 1995 | 367.17 | 1,305.20 | 20.35 | 548.91 | 22,840.33 | 274.90 | 72.19 | 335.85 | 6,779.71 | 4,264.83 | 2,074.71 | 139.04 |
| TOT. 2019 | 15,213.51 | 277,768.59 | 9,663.61 | 67,501.60 | 226,703.45 | 82,339.45 | 4,026.20 | 63,531.41 | 19,962.15 | 10,277.86 | 6,757.66 | 1,573.96 |
| TOT. 1995 | 11,501.20 | 281,271.07 | 7,560.72 | 48,487.92 | 191,109.01 | 55,537.35 | 4,193.39 | 26,330.68 | 16,250.73 | 9,433.70 | 6,665.65 | 1,821.73 |

Ref: TPP: Power Plants, MFC: Manufacturing own fuel consumption, ROC: Refinery own fuel consumption, FPR: Fuel production, FUG: Fugitive, venting and flare, ROT: Road transport, DOA. Domestic Aviation, R+N: Railroad and navigation, R+C: Residential and commercial, FAG: Fuel use in agriculture, MOP: Manufacturing own process, LF: Livestock feeding, AG: Agriculture, AWB: Agriculture waste burning, OBB. Open biomass burning.







**Table A5: Impact of COVID-19 lockdown on Argentine emissions: Summary of monthly emissions for April 2020 and April 2019**

| ACTIVITY | CO2 (Gg) | CH4 (Mg) | N2O (Mg) | NOx (Mg) | CO (Mg) | NMVOC (Mg) | SO2 (Mg) | NH3 (Mg) | TSP (Mg) | PM10 (Mg) | PM2.5 (Mg) | BC (Mg) |
|---|---|---|---|---|---|---|---|---|---|---|---|---|
| TPP 2019 | 2,540.43 | 383.39 | 108.15 | 9,010.18 | 3,160.35 | 495.75 | 23.29 | 6.24 | 45.53 | 45.38 | 39.18 | 1.47 |
| TPP 2020 | 2,294.77 | 347.09 | 97.93 | 8,141.54 | 2,849.11 | 448.23 | 17.23 | 5.63 | 40.78 | 40.71 | 35.13 | 1.19 |
| MFC 2019 | 2,093.32 | 181.19 | 24.26 | 3,514.07 | 1,505.87 | 222.09 | 231.72 | - | 831.10 | 785.91 | 739.35 | 199.83 |
| MFC 2020 | 1,798.00 | 148.91 | 19.89 | 3,034.91 | 1,289.28 | 184.70 | 192.49 | - | 675.23 | 638.86 | 601.50 | 162.11 |
| ROC 2019 | 489.19 | 10.30 | 1.14 | 1,306.22 | 212.32 | 47.06 | 42.53 | - | 13.28 | 10.20 | 7.00 | 1.13 |
| ROC 2020 | 377.30 | 7.39 | 0.81 | 1,004.66 | 148.02 | 35.90 | 20.31 | - | 9.35 | 7.11 | 5.57 | 0.82 |
| FPR 2019 | 1,012.72 | 63.64 | 7.50 | 318.97 | 677.88 | 24,051.49 | 524.50 | - | 1.20 | 1.20 | 1.20 | 0.28 |
| FPR 2020 | 790.18 | 45.94 | 7.36 | 313.48 | 557.55 | 23,501.96 | 519.32 | - | 0.91 | 0.91 | 0.91 | 0.21 |
| FUG 2019 | 286.31 | 38,218.88 | 5.58 | 0.00 | 7.79 | 13.11 | 0.01 | 0.02 | 0.21 | 0.22 | 0.21 | 0.06 |
| FUG 2020 | 203.68 | 34,077.40 | 4.99 | 0.00 | 7.11 | 9.83 | 0.01 | 0.02 | 0.19 | 0.20 | 0.19 | 0.05 |
| ROT 2019 | 4,041.20 | 1,247.94 | 296.17 | 34,981.68 | 160,653.89 | 32,511.94 | 1,119.41 | 1,070.96 | 1,253.78 | 1,003.03 | 902.72 | 240.59 |
| ROT 2020 | 2,258.54 | 496.35 | 131.63 | 16,620.72 | 60,076.25 | 12,352.88 | 796.19 | 467.55 | 957.93 | 766.35 | 689.71 | 184.77 |
| DOA 2019 | 150.08 | 1.05 | 4.20 | 524.77 | 209.91 | 104.95 | 95.19 | 1.87 | 1.68 | 1.05 | 0.05 | 0.15 |
| DOA 2020 | - | - | - | - | - | - | - | - | - | - | - | - |
| R+N 2019 | 115.04 | 10.49 | 2.93 | 296.29 | 2,700.24 | 0.27 | 113.80 | 641.24 | 191.46 | 190.85 | 172.91 | 112.39 |
| R+N 2020 | 76.93 | 6.89 | 1.96 | 184.70 | 1,915.03 | 0.18 | 68.16 | 477.60 | 147.06 | 146.98 | 132.83 | 86.34 |
| R+C 2019 | 2,390.18 | 42.58 | 4.26 | 2,554.52 | 1,277.26 | 85.15 | 12.77 | 12.77 | 51.09 | 51.09 | 51.09 | 5.11 |
| R+C 2020 | 1,579.79 | 184.38 | 6.87 | 1,616.78 | 7,176.24 | 558.13 | 118.59 | 6.39 | 543.21 | 508.55 | 495.14 | 38.26 |
| FAG 2019 | 1,062.98 | 42.52 | 24.15 | 4,655.79 | 3,927.90 | 657.45 | 58.35 | 12.75 | 570.23 | 509.86 | 509.86 | 234.80 |
| FAG 2020 | 1,198.40 | 47.94 | 27.23 | 5,248.89 | 4,428.28 | 741.20 | 65.79 | 14.37 | 642.88 | 574.81 | 574.81 | 264.71 |
| MOP 2019 | 2,540.43 | 383.39 | 108.15 | 9,010.18 | 3,160.35 | 495.75 | 23.29 | 6.24 | 45.53 | 45.38 | 39.18 | 1.47 |
| MOP 2020 | 2,294.77 | 347.09 | 97.93 | 8,141.54 | 2,849.11 | 448.23 | 17.23 | 5.63 | 40.78 | 40.71 | 35.13 | 1.19 |
| TOT. 2019 | 14,181.46 | 40,201.98 | 478.34 | 57,162.51 | 174,333.41 | 58,189.27 | 2,221.58 | 1,745.85 | 2,959.58 | 2,598.78 | 2,423.58 | 795.81 |
| TOT. 2020 | 10,577.58 | 35,362.28 | 298.67 | 36,165.69 | 78,446.87 | 37,833.00 | 1,798.10 | 971.55 | 3,017.55 | 2,684.48 | 2,535.79 | 738.47 |
| (20-19)/19 | -34.1% | -13.7% | -60.2% | -58.1% | -122.2% | -53.8% | -23.6% | -79.7% | 1.9% | 3.2% | 4.4% | -7.8% |

Ref: TPP: Power Plants, MFC: Manufacturing own fuel consumption, ROC: Refinery own consumption, FPR: Fuel production, FUG: Fugitive, venting and flare, ROT: Road transport, DOA. Domestic Aviation, R+N: Railroad and navigation, R+C: Residential and commercial, FAG: Fuel use in agriculture, MOP: Manufacturing own process, LF: Livestock feeding, AG: Agriculture, AWB: Agriculture waste burning, OBB: Open biomass burning.





**Table A6: TCNA inventory: annual GHG emissions (CO2eq) for Argentina**

| Year | Thermal power plants | Industry Own generation | Refineries Own consumption | Oil and gas wells Fuel production | Fugitive | Transport road | aviation | RR+Nav | Residential R+C+G | Industry process | Livestock | Agriculture | AWB | Open Fire | TOTAL CO2eq |
|---|---|---|---|---|---|---|---|---|---|---|---|---|---|---|---|
|  | 1A | 1A2 | 1A1b | 1A1c | 1B2 | 1A3b | 1A3a | 1A3c-d | 1A4a-b | 2B-2C | 3A | 3C | 3C | 4D |  |
| 1990 | 15,706.88 | 14,669.44 | 5,821.28 | 3,447.89 | 6,950.76 | 25,507.58 | 815.39 | 288.37 | 24,517.72 | 9,540.84 | 87,636.74 | 349.19 | 4,887.05 | 11,169.89 | 211,309.03 |
| 1991 | 19,136.44 | 14,856.82 | 6,009.10 | 4,892.44 | 7,408.33 | 29,461.89 | 733.85 | 330.67 | 24,720.74 | 8,378.34 | 88,594.13 | 463.43 | 4,938.48 | 11,271.16 | 221,195.81 |
| 1992 | 18,017.77 | 15,463.15 | 6,965.58 | 3,694.22 | 7,750.94 | 32,019.02 | 884.85 | 328.63 | 25,140.64 | 8,303.30 | 89,722.18 | 529.82 | 4,512.90 | 11,342.06 | 224,675.07 |
| 1993 | 18,015.32 | 15,530.52 | 6,814.21 | 3,474.92 | 8,309.04 | 32,737.29 | 948.27 | 344.06 | 26,223.75 | 8,912.40 | 90,799.21 | 1,282.76 | 4,949.65 | 11,443.96 | 229,785.36 |
| 1994 | 17,628.19 | 16,829.50 | 5,282.66 | 3,740.68 | 8,866.12 | 35,737.92 | 1,951.31 | 363.93 | 26,742.26 | 9,721.20 | 91,952.85 | 1,883.75 | 5,420.74 | 7,415.99 | 233,537.10 |
| 1995 | 18,166.10 | 17,282.35 | 5,022.54 | 4,080.22 | 9,564.93 | 36,945.09 | 1,514.86 | 338.02 | 27,148.36 | 9,328.91 | 89,756.38 | 2,105.59 | 5,085.33 | 7,669.22 | 234,007.89 |
| 1996 | 21,285.91 | 19,657.44 | 4,438.59 | 5,085.91 | 10,516.06 | 39,232.40 | 1,314.52 | 661.29 | 28,071.42 | 9,836.97 | 88,821.63 | 3,248.31 | 6,276.31 | 7,163.02 | 245,609.78 |
| 1997 | 19,134.48 | 20,276.39 | 4,917.95 | 6,910.75 | 11,067.24 | 41,133.64 | 1,250.39 | 610.85 | 28,671.85 | 10,826.80 | 87,426.72 | 3,150.95 | 6,498.68 | 5,200.40 | 247,077.10 |
| 1998 | 21,058.34 | 20,789.27 | 4,626.76 | 8,668.25 | 11,319.03 | 41,052.62 | 1,454.38 | 660.72 | 29,365.26 | 10,418.14 | 86,637.43 | 3,276.85 | 5,419.29 | 6,473.43 | 251,219.77 |
| 1999 | 25,361.58 | 19,606.88 | 4,260.68 | 6,853.12 | 11,751.22 | 40,063.34 | 1,625.74 | 525.97 | 30,813.07 | 10,039.09 | 87,100.90 | 3,902.55 | 5,706.53 | 5,087.66 | 252,698.34 |
| 2000 | 24,930.20 | 19,644.42 | 4,102.38 | 7,270.08 | 12,002.19 | 42,946.45 | 1,456.41 | 554.78 | 31,740.68 | 10,885.59 | 90,383.24 | 3,801.71 | 6,473.61 | 11,855.40 | 268,047.17 |
| 2001 | 18,588.23 | 19,024.53 | 3,897.30 | 7,466.04 | 12,324.69 | 39,290.91 | 1,221.01 | 537.51 | 32,065.79 | 10,576.84 | 92,194.44 | 4,001.92 | 6,472.31 | 16,481.77 | 264,143.29 |
| 2002 | 15,629.79 | 19,627.82 | 4,175.28 | 7,869.93 | 11,878.26 | 36,005.43 | 1,051.15 | 367.43 | 30,385.11 | 11,208.32 | 97,328.20 | 3,775.15 | 6,552.59 | 10,447.44 | 256,301.92 |
| 2003 | 19,294.77 | 20,699.18 | 4,589.06 | 8,040.06 | 12,695.49 | 36,180.78 | 993.08 | 413.59 | 31,773.64 | 12,198.88 | 103,077.81 | 4,886.99 | 6,613.71 | 11,451.45 | 272,908.48 |
| 2004 | 24,327.20 | 21,513.38 | 4,427.33 | 8,478.70 | 12,913.57 | 39,735.19 | 1,129.51 | 488.02 | 34,189.58 | 13,146.01 | 105,890.70 | 5,634.71 | 7,464.25 | 4,966.31 | 284,304.46 |
| 2005 | 26,647.44 | 21,648.02 | 3,956.11 | 8,123.95 | 12,774.80 | 41,411.57 | 1,154.19 | 528.46 | 37,339.45 | 14,491.42 | 106,500.77 | 5,336.95 | 7,389.84 | 5,947.75 | 293,250.72 |
| 2006 | 29,569.33 | 23,149.73 | 4,347.12 | 8,182.17 | 12,910.18 | 44,517.82 | 1,051.50 | 609.38 | 38,947.71 | 15,127.06 | 108,307.50 | 6,397.94 | 8,222.43 | 5,548.83 | 306,888.71 |
| 2007 | 34,148.97 | 22,814.74 | 4,804.72 | 8,977.27 | 12,887.55 | 47,496.82 | 1,113.14 | 418.64 | 43,609.29 | 15,764.48 | 108,912.19 | 7,209.60 | 8,248.23 | 4,828.97 | 321,234.62 |
| 2008 | 37,551.54 | 23,497.17 | 5,181.20 | 9,757.38 | 12,828.71 | 48,113.19 | 1,227.32 | 403.06 | 41,330.10 | 15,117.25 | 105,199.48 | 5,242.94 | 6,316.70 | 5,579.43 | 317,345.47 |
| 2009 | 34,574.48 | 22,298.53 | 5,180.49 | 10,271.38 | 12,134.80 | 48,806.22 | 1,265.50 | 403.63 | 40,661.47 | 12,766.63 | 100,433.97 | 4,887.72 | 8,224.93 | 6,485.02 | 308,394.76 |
| 2010 | 37,231.26 | 22,911.27 | 5,884.67 | 10,060.11 | 11,871.86 | 49,949.26 | 1,072.06 | 1,267.85 | 41,853.22 | 15,038.69 | 67,294.02 | 6,567.54 | 9,040.13 | 5,202.85 | 285,244.80 |
| 2011 | 42,719.05 | 23,786.20 | 5,423.89 | 9,978.06 | 11,785.01 | 51,675.56 | 1,029.39 | 1,672.33 | 42,581.64 | 16,209.16 | 68,960.22 | 7,136.69 | 7,786.69 | 4,398.59 | 295,142.48 |
| 2012 | 45,839.43 | 21,881.67 | 5,541.97 | 10,015.44 | 11,492.12 | 49,547.25 | 1,123.33 | 1,619.72 | 42,563.09 | 15,384.33 | 72,408.78 | 6,109.88 | 8,628.73 | 3,525.62 | 295,681.35 |
| 2013 | 45,387.65 | 21,873.91 | 5,874.32 | 10,002.27 | 11,146.36 | 52,200.96 | 1,425.95 | 1,264.30 | 44,474.53 | 16,378.75 | 74,069.66 | 6,540.19 | 11,686.42 | 3,609.97 | 305,935.25 |
| 2014 | 42,862.29 | 20,911.32 | 5,384.70 | 10,093.15 | 11,178.27 | 54,278.65 | 1,424.71 | 1,225.31 | 46,118.80 | 16,578.47 | 75,076.70 | 7,141.45 | 11,065.15 | 3,987.29 | 307,326.26 |

All values are expressed in Gigagram (Gg)








**Table A7: Comparison total annual values GEAA and TCNA from 1995 through 2015**

| SECTOR | TPP | MFC | ROC | FPR | FUG | ROT | DOA | R+N | R+C |
|---|---|---|---|---|---|---|---|---|---|
| 1995 | -0.7% | -7.6% | -28.3% | 16.8% | -9.8% | 11.3% | 2.9% | 20.6% | 9.4% |
| 1996 | -0.9% | -7.2% | -12.9% | 2.7% | -15.0% | 6.4% | 4.0% | -40.0% | 8.8% |
| 1997 | 2.0% | -6.6% | 4.1% | -19.1% | -8.8% | 3.4% | 3.8% | -29.1% | 5.0% |
| 1998 | -0.4% | -0.1% | 17.4% | -27.4% | -1.6% | 6.7% | 3.2% | -30.0% | 2.7% |
| 1999 | 0.4% | -1.1% | 2.3% | -7.1% | -6.7% | 4.9% | 2.6% | -11.7% | 5.1% |
| 2000 | 1.0% | -3.9% | -6.3% | -8.3% | -9.0% | -4.9% | 2.7% | -18.3% | 4.9% |
| 2001 | 1.5% | -7.5% | 1.8% | -7.2% | -5.9% | -7.7% | 1.0% | 13.2% | -1.9% |
| 2002 | 1.6% | 3.7% | 11.1% | -9.4% | -1.0% | -2.5% | 1.0% | 14.7% | -3.6% |
| 2003 | -6.5% | -2.7% | 13.4% | -7.2% | 2.2% | -2.3% | 1.0% | 28.5% | 1.7% |
| 2004 | -0.2% | -7.4% | 0.5% | -13.7% | -2.6% | 1.4% | 1.0% | 42.7% | 3.9% |
| 2005 | 1.9% | -3.1% | 4.3% | -7.3% | -7.2% | -6.2% | 1.0% | 39.9% | -1.9% |
| 2006 | -2.1% | -8.6% | -0.4% | -4.5% | -7.4% | -5.9% | 1.0% | 38.0% | -2.7% |
| 2007 | -3.5% | -14.0% | -6.2% | -4.2% | -5.8% | -1.6% | 1.0% | 26.2% | -11.4% |
| 2008 | -0.7% | -3.7% | -15.5% | -5.3% | -6.2% | 3.5% | 1.0% | 29.1% | -7.5% |
| 2009 | -0.7% | -9.3% | -0.8% | -5.1% | -3.5% | -5.4% | 1.0% | 24.0% | 0.7% |
| 2010 | -3.0% | -1.2% | 2.5% | -7.7% | -3.8% | -6.1% | 27.5% | 27.4% | -2.9% |
| 2011 | -5.3% | -3.6% | 5.5% | -7.4% | -6.6% | -4.8% | 30.6% | 22.4% | -9.3% |
| 2012 | -4.8% | -8.1% | 9.7% | -6.9% | -5.3% | 0.0% | 21.9% | 19.5% | -5.4% |
| 2013 | -5.5% | -8.5% | 3.4% | -7.0% | -2.8% | -0.1% | 0.1% | 22.9% | 0.9% |
| 2014 | 2.1% | -19.0% | 11.2% | -4.3% | -6.2% | -5.7% | 3.8% | 26.0% | -7.1% |
| Average | -1.19% | -5.98% | 0.83% | -6.97% | -5.65% | -0.79% | 5.59% | 13.30% | -0.53% |

op

| SECTOR | MOP | LF | AG | AWB | OBB | Total | Std. Dev |
|---|---|---|---|---|---|---|---|
| 1995 | -7.5% | 12.3% | -21.6% | -37.8% | -25.2% | 5.5% | 18% |
| 1996 | 3.7% | 11.3% | -23.2% | -36.8% | -18.8% | 4.3% | 17% |
| 1997 | -10.2% | 8.2% | -19.0% | -27.9% | 19.1% | 2.7% | 14% |
| 1998 | 2.5% | 9.9% | -28.4% | -12.8% | -17.3% | 3.8% | 15% |
| 1999 | 6.9% | 14.1% | -12.4% | -14.5% | 6.4% | 6.3% | 9% |
| 2000 | 8.3% | 7.9% | -8.2% | -15.6% | -74.5% | -0.5% | 21% |
| 2001 | 10.0% | 7.3% | -13.1% | -7.4% | -26.5% | -0.8% | 10% |
| 2002 | 11.3% | -0.2% | -6.0% | -9.6% | -17.1% | -0.8% | 9% |
| 2003 | -0.4% | -5.7% | -12.6% | 1.8% | -6.9% | -3.3% | 10% |
| 2004 | 4.5% | -0.8% | -11.7% | -1.0% | 42.1% | 0.3% | 17% |
| 2005 | 0.2% | -6.3% | -13.7% | -0.5% | 8.0% | -3.9% | 13% |
| 2006 | 4.6% | -7.6% | -15.6% | 12.0% | 24.7% | -4.7% | 14% |
| 2007 | 1.0% | -11.3% | -9.1% | 16.6% | 17.9% | -7.4% | 12% |
| 2008 | 2.3% | -6.6% | -7.1% | 17.4% | 52.3% | -2.3% | 18% |
| 2009 | 1.9% | -11.6% | -2.3% | 9.4% | 7.7% | -5.4% | 9% |
| 2010 | -0.2% | 22.9% | 0.7% | -0.4% | -2.3% | 4.1% | 12% |
| 2011 | -0.1% | 21.8% | -15.4% | 4.0% | 9.6% | 2.3% | 14% |
| 2012 | 2.6% | 14.4% | 3.9% | 2.2% | 35.9% | 2.5% | 13% |
| 2013 | 0.2% | 13.7% | -16.2% | -19.9% | 55.9% | 2.8% | 19% |
| 2014 | 1.0% | 20.4% | -19.2% | -9.6% | -9.7% | 2.2% | 13% |
| Average | 2.13% | 5.71% | -12.51% | -6.52% | 4.07% | 0.40% | 13.82% |

- The percentage difference has been computed as (GEAA – TCNA) / GEAA * 100.%

Ref: TPP (1A1): Power Plants, MFC (1A2): Manufacturing own fuel consumption, ROC (1A1b): Refinery own consumption, FPR (1A1c): Fuel production, FUG (1B2): Fugitive, venting and flare, ROT (1A3b): Road transport, DOA(1A3a). Domestic Aviation, R+N (1A3c-d): Railroad and navigation, R+C (NG) (1A4a-b): Residential and commercial, MOP (2B-2C): Manufacturing own process, LF (3A): Livestock feeding, AG (3C): Agriculture, AWB: Agriculture waste burning, OBB (4D). Open biomass burning.






**Table A8: Comparison total annual values GEAA and EDGAR from 1995 through 2015**

| 1995-2015 | GEAA-EDGAR | CO$_2$ | | CH$_4$ | | N$_2$O | | CO$_{2eq}$ | |
|---|---|---|---|---|---|---|---|---|---|
| Stat./ sector | | Mean | Std. Dev. | Mean | Std. Dev. | Mean | Std. Dev. | Mean | Std. Dev. |
| TPP | 1A1a | -23.8% | 4.9% | 78.7% | 14.9% | 139.9% | 15.4% | -22.3% | 4.6% |
| MFC | 1A2 | -1.9% | 14.8% | 12.3% | 20.7% | -5.6% | 17.5% | -35.5% | 10.4% |
| ROC/FPR | 1A1bc | -17.5% | 8.2% | -99.5% | 6.6% | -30.9% | 19.3% | -18.6% | 8.2% |
| | 1B1 | | | -148.5% | 32.5% | | | -172.3% | 21.0% |
| FUG | 1B2 | 78.9% | 62.2% | -92.9% | 15.3% | 174.9% | 5.0% | -71.6% | 19.0% |
| ROT | 1A3b | 14.5% | 5.9% | 1.8% | 3.8% | 96.4% | 8.1% | 14.4% | 5.4% |
| DOA | 1A3a | 3.1% | 10.1% | 4.5% | 9.4% | 4.5% | 9.4% | 4.6% | 9.4% |
| R+N | 1A3c-d | 37.7% | 32.6% | 40.2% | 33.5% | 36.6% | 33.8% | 40.1% | 34.1% |
| R+C | 1A4a-b | -25.9% | 6.6% | -83.7% | 9.8% | -165.1% | 3.1% | -28.1% | 5.7% |
| **Energy** | | **-7.7%** | **3.2%** | **-91.2%** | **14.5%** | **-14.7%** | **11.4%** | **-21.8%** | **3.5%** |
| MOP | 2B-2C | 43.2% | 13.3% | -87.0% | 34.1% | 43.1% | 14.0% | 45.7% | 11.4% |
| LF | 3A | | | -22.7% | 8.8% | | | 5.9% | 7.8% |
| AG | 3C7 | -177.4% | 4.9% | 21.3% | 31.5% | -113.6% | 21.2% | -162.7% | 6.6% |
| AWB | 3C | | | -171.3% | 8.0% | -176.8% | 6.5% | -172.6% | 7.6% |
| OBB | 4D | | | -175.5% | 10.6% | | | -157.3% | 17.6% |
| **Total** | | **-7.1%** | **3.7%** | **-69.7%** | **14.5%** | **0.8%** | **6.1%** | **-36.0%** | **6.4%** |

| 1995-2015 | GEAA-EDGAR | NO$_x$ | | CO | | PM$_{10}$ | | PM$_{2.5}$ | |
|---|---|---|---|---|---|---|---|---|---|
| Stat./ sector | | Mean | Std. Dev. | Mean | Std. Dev. | Mean | Std. Dev. | Mean | Std. Dev. |
| TPP | 1A1a | 14.2% | 10.2% | 5.3% | 7.2% | -102.5% | 27.6% | -72.9% | 29.5% |
| MFC | 1A2 | -96.9% | 7.7% | -149.5% | 28.5% | -87.5% | 18.7% | -61.9% | 19.9% |
| ROC/FPR | 1A1bc | 105.0% | 24.5% | 124.2% | 7.5% | 30.5% | 33.8% | 38.1% | 35.6% |
| | 1B1 | | | | | | | | |
| FUG | 1B2 | -84.3% | 40.3% | -152.5% | 40.8% | -28.4% | 58.0% | -22.2% | 59.0% |
| ROT | 1A3b | 1.6% | 12.6% | -12.6% | 11.2% | -5.8% | 9.3% | -16.3% | 9.2% |
| DOA | 1A3a | 7.1% | 9.4% | 25.8% | 8.8% | -157.6% | 3.8% | -197.9% | 0.2% |
| R+N | 1A3c-d | 26.3% | 35.4% | -45.1% | 50.1% | -77.6% | 31.1% | -46.4% | 107.1% |
| R+C | 1A4a-b | 39.7% | 4.3% | 11.2% | 20.0% | -35.2% | 35.0% | 8.9% | 35.0% |
| **Energy** | | **-4.7%** | **9.0%** | **-19.3%** | **10.8%** | **-76.8%** | **6.0%** | **-40.4%** | **7.7%** |
| MOP | 2B-2C | 88.7% | 9.1% | -82.8% | 19.6% | -68.9% | 16.9% | -48.7% | 17.7% |
| LF | 3A | 89.4% | 8.4% | | | 102.4% | 5.4% | 161.0% | 4.2% |
| AG | 3C | -30.4% | 35.3% | | | 126.0% | 9.5% | -193.9% | 0.9% |
| AWB | 3C | | | | | | | | |
| OBB | 4D | -136.1% | 23.5% | | | -95.3% | 40.4% | -134.5% | 27.5% |
| **Total** | | **-11.0%** | **7.8%** | **-60.9%** | **9.2%** | **-67.6%** | **18.9%** | **-109.1%** | **16.9%** |


- The percentage difference has been computed as (GEAA – EDGAR) / GEAA * 100.%

Ref: PP: Power Plants, MFC: Manufacturing own fuel consumption, ROC: Refinery own consumption, FPR: Fuel production, FUG: Fugitive, venting and flare, ROT: Road transport, DOA. Domestic Aviation, R+N: Railroad and navigation, R+C (NG): Residential and commercial (natural gas), R+C (OF) Residential and commercial (other fuels), FAG: Fuel use in agriculture, MOP: Manufacturing own process, LF: Livestock feeding, AG: Agriculture, AWB: Agriculture waste burning, OBB. Open biomass burning.


**FIGURES**

a)

b)

c)

d)

**Figure A1: Calculated VKT for gasoline vehicles; b) Calculated VKT for gasoline vehicles at central area of Argentina. c) Monthly**
**fuel sales: Gasoline blue line); Gas oil (red line); Compressed natural gas (CNG) (black line); d) Monthly emissions (in Mg) from**
**road transport between January-1995 through April 2020; CO (blue line) and NOx (black line) left axis, PM₁₀ (red line) right axis.**








a)

b)

c)

d)

e)

f)

**Figure A2. a) Monthly NOx and SO₂ emissions (Mg) from thermal power plants; b) average seasonal NOx and SO₂ emissions 1995-2019 (Mg) from thermal power plants; c) Monthly oil (m³) and gas production (1000 m³); d) Monthly methane emissions (Mg) from fuel production. e) Monthly aerokerosene sales at airports (m³) for domestic and international flights; f) Monthly CO and NOx emissions from aviation.**

a)

b)

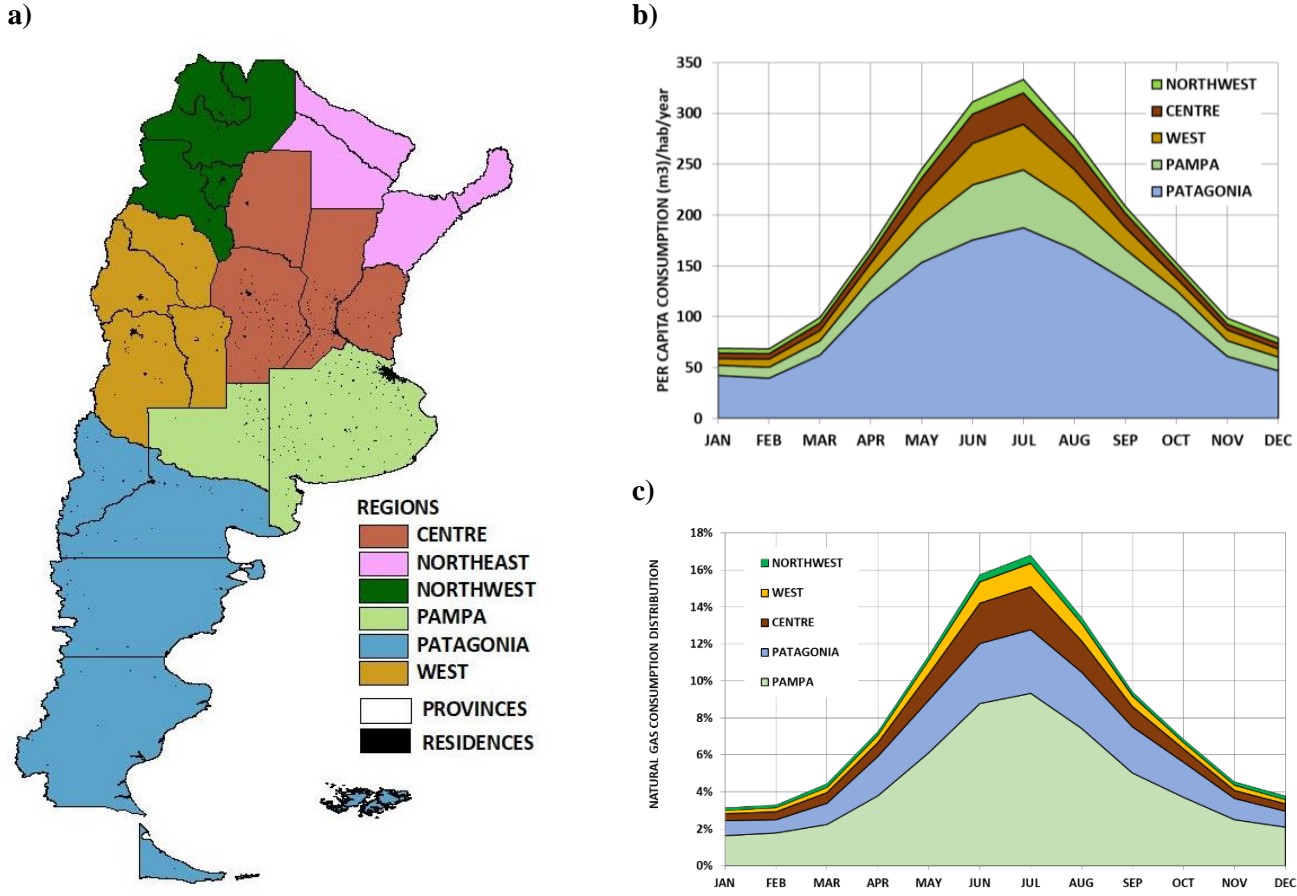

c)

**Figure A3. a) Regions and provinces with natural gas consumption at homes, b) Per capita annual natural gas consumptions, c) regional and seasonal distribution of natural gas consumptions per region (% of total annual consumption).**






a)

b)

c)


**Figure A4. a) Railroad network and navigation ports, b) seasonal railroad freight (Million t. per km) and passenger activity (Million passengers per km), c) Monthly railroad activity and fuel consumption (m³) and passenger activity (Million passengers per km).**





a)

b)

c)

d)

**Figure A5. a) Land types for Argentina; b) monthly average precipitation (mm/cell); c) monthly average burned area (ha/cell); d) PM2.5 emissions in (kg/cell) for Sept. 2017.**


a)

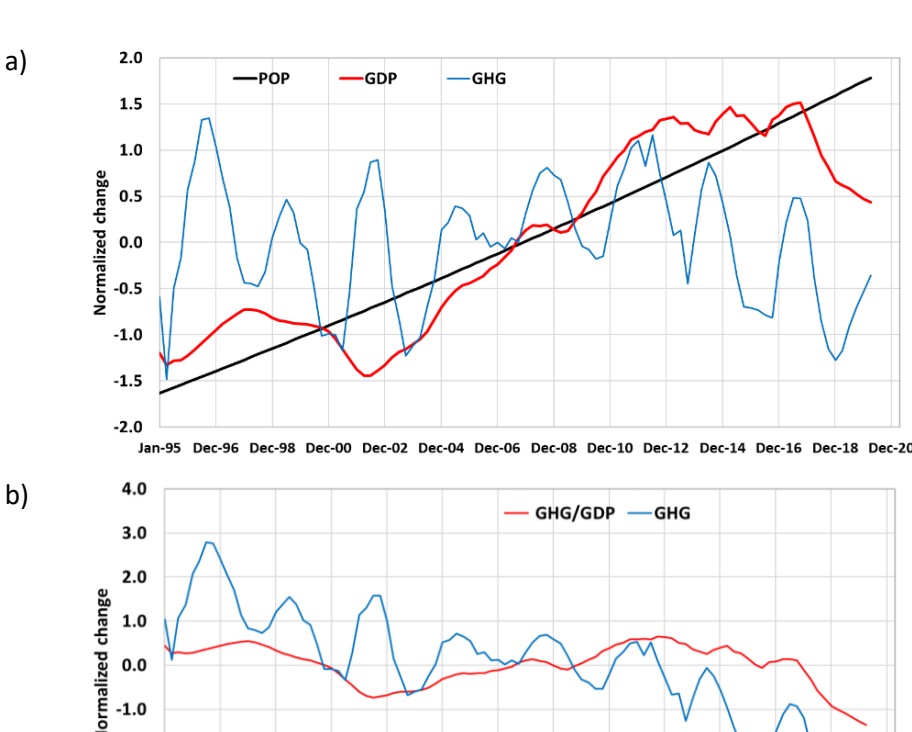

b)

c)

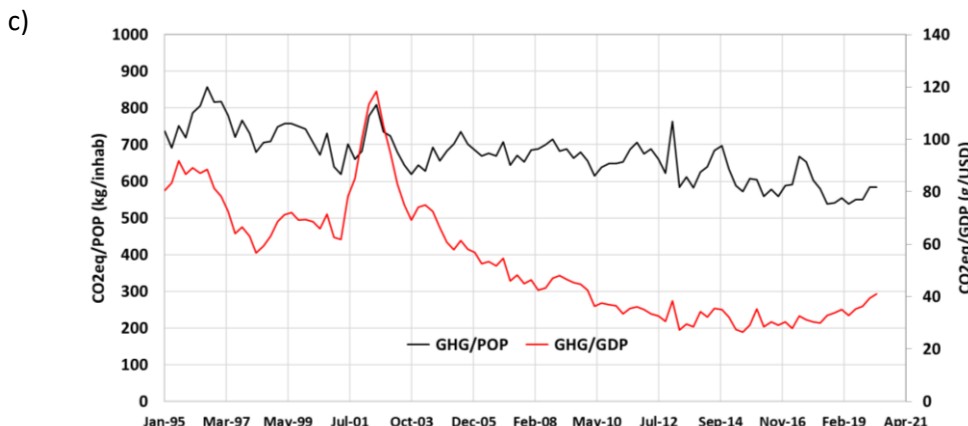

**Figure A6: Normalized Change in a) Population, Gross Domestic Product and GHG in terms of CO2eq between 1995 and 2020; b) Population de-trended GDP and GHG. c) De-trended GHG/cap and GHG/GDP.** The normalized function is obtained by subtracting the function mean value and divided by its standard deviation.


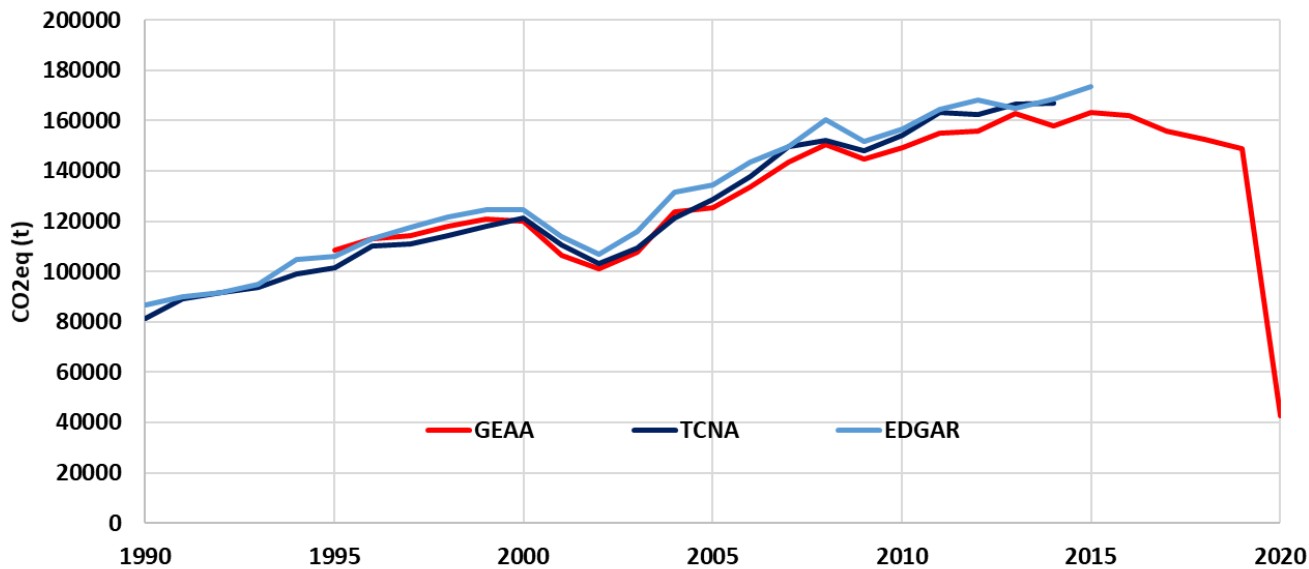

**Figure A7. Comparison of annual GHG emissions for the energy sector excluding refining and fugitive emissions from fuel production (1A1b, 1B1, 1B2) between the different inventories considered in this work. Note that EDGAR has 2.5 times more CH4 emissions from the 3 removed sectors than GEAA and TCNA (see Table A8 Append.).**
