# Peer review of "High resolution seasonal and decadal inventory of anthropic gasphase and particle emissions for Argentina"

_Earth System Science Data, 2021_

## Author Comment (AC1)

**Response to referee comments 1 (RC1)**

Dear Referee 1,

Thank you for the revision of our manuscript number ESSD-2021-81. We have reviewed and adjusted the manuscript considering all your observations and commentaries. Below, you will find our detailed responses. Within this response letter the following style is used: the original general comments made by the reviewer are kept in normal text (initiated with R), our responses are *in blue italics initiating with A* (Authors). We use *italics black* for other authors texts (citations, initiated with C). The corresponding edit in the manuscript are included in red.

**Referee comments 1 (RC1)**

Comment on essd-2021-81 Anonymous Referee #1 Referee comment on "High resolution seasonal and decadal inventory of anthropic gas phase and particle emissions for Argentina" by S. Enrique Puliafito et al., Earth Syst. Sci. Data Discuss., https://doi.org/10.5194/essd-2021-81-RC1, 2021

Review of "High resolution seasonal and decadal inventory of anthropic gas phase and particle emissions for Argentina" ESSD-2021-81.

R1: This is an excellent study and deserves to be published. In fact, it has been difficult to find fault with it beyond errors in language that will be caught by Copernicus' copy-editing stage. It is clear this paper builds on work that has been undertaken over several years, resulting in several other papers since at least 2015. The paper under review brings together these works into a comprehensive study covering all sectors and a range of emission species. The comparison with EDGAR is particularly useful, as EDGAR is widely used but uses a relatively standard method across all countries.

*A. Thank you very much for your comments and recommendations, which helped to improve our study.*

R1. I have a few very minor comments.

R1. Line 148: National Communications are submitted to the UNFCCC, not the IPCC.

*A. Yes, you are wright. Corrected.*

R1. Line 633: The authors state that this is 'clearly' a result of EDGAR using a low resolution population map. Can they support this statement with reference to EDGAR publications? I think the more information that the EDGAR team has about how to make improvements, the better.

*A. Analyzing the spatial distribution of the EDGAR emissions one can distinguish the inner border of less populated provinces as well as districts borders in more populated areas. The use of country and subnational borders to distribute population data is confirmed in Janssens-Maenhout, G. et al, (2019) by means of the Gridded Population of the World map (GPWv3). According to Crippa et al, (2020), and more specifically Janssens-Maenhout, G. et al, (2019):*

> C:"*…gridded world population* (are) *provided by the Center for International Earth Science Information Network (CIESIN, 2005 and updated in 2011) for the years 1990, 1995, 2000, 2005, 2010 and 5 projected to 2015. In-house proxy datasets are developed by dividing the total population into rural and urban. These data are applied in order to cover the country area and population and take into account the fraction of country data in cells with an intersection of the country's borders*…." (Suppl. Mat., Page 14). EDGAR uses Gridded Population of the World, Version Three (GPWv3) from CIESIN, which* "*…is constructed from national or subnational input units (usually administrative units) of varying resolutions…" (CIESIN, 2005, GPWv3).*

*C:"…Gridded Population of the World, Version Three (GPWv3): This is a gridded, or raster, data product that renders global population data at the scale and extent required to demonstrate the spatial relationship of human populations and the environment across the globe. The purpose of GPW is to provide a spatially disaggregated population layer that is compatible with data sets from social, economic, and Earth science fields. **The gridded data set is constructed from national or subnational input units (usually administrative units) of varying resolutions**. The native grid cell resolution is 2.5 arc-minutes, or ~5km at the equator, although aggregates at coarser resolutions are also provided. Separate grids are available for population count and density per grid cell…"* ([https://sedac.ciesin.columbia.edu/data/collection/grump-v1/about-us](https://sedac.ciesin.columbia.edu/data/collection/grump-v1/about-us))

*C:…"Global Rural-Urban Mapping Project, Version One (GRUMPv1) This project builds on GPW to construct a common geo-referenced framework of urban and rural areas by combining census data with satellite data. GRUMPv1 actually comprises three data products. First, GRUMPv1 provides a higher resolution gridded population data product at 30 arc-seconds, or ~1km at the equator, for 1990, 1995, and 2000. Second, GRUMPv1's urban extents data set delineates urban areas based on NOAA's night-time lights data set and buffered settlement centroids (where night lights are not sufficiently bright). Third, GRUMPv1 provides a points data set of all urban areas with populations of greater than 1,000 persons, which may be downloaded in Excel, CSV, and shapefile formats. As with GPW, there is an extensive map collection depicting the data sets at country, continental, and global levels".* ([https://sedac.ciesin.columbia.edu/data/collection/grump-v1/about-us](https://sedac.ciesin.columbia.edu/data/collection/grump-v1/about-us)).

Crippa, M., Solazzo, E., Huang, G. *et al.* High resolution temporal profiles in the Emissions Database for Global Atmospheric Research. *Sci Data* **7,** 121 (2020). https://doi.org/10.1038/s41597-020-0462-2

Janssens-Maenhout, G. *et al*. EDGAR v4.3.2 Global Atlas of the three major greenhouse gas emissions for the period 1970–2012. *Earth Syst. Sci. Data* **11**, 959–1002, https://doi.org/10.5194/essd-11-959-2019 (2019); and Supplement of Earth Syst. Sci. Data, 11, 959–1002, 2019 https://doi.org/10.5194/essd-11-959-2019-supplement

CIESIN: Gridded population of the world, version 3 (GPWv3), Center for International Earth Science INformation Network (CIESIN), USA, downloaded from http://sedac.ciesin.columbia.edu/data/collection/gpwv3, 2005 and updated in 2011.

Crippa, M., Guizzardi, D., Muntean, M., Schaaf, E., Dentener, F., van Aardenne, J. A., Monni, S., Doering, U., Olivier, J. G. J., Pagliari, V., and Janssens-Maenhout, G.: Gridded emissions of air pollutants for the period 1970–2012 within EDGAR v4.3.2, Earth Syst. Sci. Data, 10, 1987–2013, https://doi.org/10.5194/essd-10-1987-2018, 2018.

*Lines 633-635 former manuscript says:*

"The larger EDGAR emissions (negative values) for the whole district are clearly an overestimation due to not considering a high-resolution population density map, as there are no direct sources on most of this region, and most of the emissions are located on a unique location on the east-edge of the district (see red dot)".

*Line 633-635 will now be:*

According to Janssens-Maenhout et al, (2019), EDGAR uses national and subnational administrative units as proxy population data using Gridded Population of the World, Version Three (GPWv3) provided by the Center for International Earth Science Information Network (CIESIN, 2005). This approach produces an emission overestimation over many low populated regions compared to the high-resolution population density map used in GEAA.

R1: Line 637: Regarding the possible overestimation of residential emissions in EDGAR, I believe EDGAR estimates these using bioenergy data from IEA. The authors might consider checking this and adding detail

here, since potentially the IEA's bioenergy estimates for Argentina are incorrect, and this has wider consequences.

A. *According to Janssens-Maenhout, et al., (2019) EDGAR v4.3.2 basic emissions calculations (Eq. 1 and Eq. 2, pages 962-963) use a country specific latitude, longitude mask as proxy data "… [ $f_{i,j}(lat, lon, t)$ ]… ", to spatially distribute the energy consumption of year t, for a given sector i, with technology j. For Southern Hemisphere and "RCO - Energy for Buildings sector", EDGAR uses the same GWPv3 population map as spatial proxy. As total energy consumption EDGAR uses IEA World Energy Balances 2016. As we understand, IEA uses annual National Energy Balances.*

*Analyzing the residential (+ commercial + public services) sectors energy consumption, main fuel is electricity and natural gas (see Figures A1 and A2 below, extracted from IEA https://www.iea.org/data-and-statistics/data-browser?country=ARGENTINA&fuel=Energy%20consumption&indicator=ElecConsBySector; and Figure A3a is calculated in GEAA using Argentina national energy balance. Other secondary fuels are also used as kerosene, liquified gas (LGP), charcoal, together with wood and other primaries. (Figure A3b)*

[Figure]

**Figure A1:** Natural gas consumption by sector according to IEA. Source: *https://www.iea.org/data-and-statistics/data-browser?country=ARGENTINA&fuel=Energy%20consumption&indicator=ElecConsBySector*

[Figure]

**Figure A2**: Electricity consumption by sector according to IEA. Source: *https://www.iea.org/data-and-statistics/data-browser?country=ARGENTINA&fuel=Energy%20consumption&indicator=ElecConsBySector*

[Figure]

**Figure A3:** Energy consumption (TJ) for residential+ commercial + public services (R+C+P) (see also Figure 3c in manuscript).

**Figure A4:** PM10 emissions from residential+ commercial + public services (R+C+P)

*Comparing PM10 emissions using EDGAR and GEAA (Figure A4) based on the used fuels, we observed that GEAA and EDGAR are similar from 1995-2000 but diverge afterwards. The main difference is in the amount of primary energy (wood and other primaries) considered in each inventory. As alternative calculation (see NEB -National Energy Balance- green line, Figure A4) we used the same energy consumption from Figure A3 but increasing the emission factor for wood, from 404 g/GJ (EMEP2019 Tables 3.3-3.5 section 1.A.4.b.i) to 700 g/GJ obtaining a better fit with EDGAR up to year 2005. From there on, both curves still diverge, although with a smaller discrepancy than GEAA. The main reason is the variability in kerosene and wood, being wood the highest PM10 emitter (Figure A3b) (since PM10 emissions factors are much higher than the rest). While natural gas (NG) represents (on average) 56% of residential energy, kerosene + charcoal + wood + others represent only 4% of energy. On contrast, the ratio of PM10 emission factors between wood and NG (wood/NG) is 700, while for NOx emission factors wood/NG is only 2. Then an overestimation of wood (and other primaries) in NOx emission is less visible than for PM10. We then believe that EDGAR most likely overestimates wood and charcoal consumption.*

*Figure A5 compares NOx emissions between GEAA and EDGAR as shown in the manuscript. TCNA is the third national communication of Argentina to UNFCC, which is also based on the national energy balance. The green line (NEB) considers the same residential energy consumption as GEAA and TCNA, but uses a lower NG emission factors of 71 g/GJ, and 200 g/GJ for wood (EMEP2019 Tables 3.3-3.5 section 1.A.4.b.i) instead of 150 g/GJ for NG and 110 g/GJ for wood used in TCNA. We adopted in GEAA same emission factors as TCNA. There is no estimation of PM10 in TCNA.*

*In conclusion if we would have adopted (EMEP2019 Table 3.3 section 1.A.4.b.i) emissions factors as we expect EDGAR does, we would have seen an overestimation also in NOX emissions.*

[Figure]

**Figure A5:** NOx emissions from residential + commercial + public services (R+C+P)

*Line 635-644 former manuscript says:*

*"When appreciating the annual values, the differences of PM$_{10}$ (and other pollutants), show similar values between the years 1995-2000, but thereafter diverges. This difference arises from a possible overestimation on the EDGAR inventory on the amount of firewood and charcoal used for heating and cooking in homes. In effect, this amount has been decreasing significantly since 2002, being replaced by an increase in the use of natural gas and LPG (Figure 3c); therefore, EDGAR trends should be corrected (Figure 8d). For estimating the residential emissions, as mention in Section 2.3.4 GEAA uses the census fractions map (INDEC, 2020) which gives fine detail of the location and amount of homes, specifying the main fuel used for cooking and heating (natural gas, wood, etc.). For NOx (Figure 9d) EDGAR overestimates GEAA values, which is seen as mostly blue areas (negative values) in Figure 9b. Since both annual series show equivalent variations, it is most probably that the discrepancies arise from the use of different emissions factors in each inventory. "*

*Line 635-644 will be rephrased as:*

*When appreciating the annual values, the differences of $PM_{10}$ (and other pollutants), show similar values between the years 1995-2000, but thereafter diverge. Firewood, charcoal and other primary energy sources used for heating and cooking in homes have been very variable but with decreasing trend since 2003, being replaced by increasing use of natural gas and LPG (Figure 3c). While natural gas (NG) represents (on average) 56% of residential energy, kerosene + charcoal + wood + other primaries represent only 4% of energy consumption at households. However, the ratio of $PM_{10}$ emission factors between wood and NG (wood/NG) is 600 to 700, while for NOx emission factors the wood/NG ratio is only 1.2 to 2. Then, any overestimation of wood (and other primaries) will be more visible in $PM_{10}$ emissions (Figure 8d) than for NOx (Figure 9d). As energy consumption inputs, EDGAR uses the International Energy Agency (IEA) World Energy Balances 2016 (Janssens-Maenhout, et al., 2019), however wood and other primary energy inputs may have been overestimated, given the high variability, or they might have used a constant per capita consumption. The 40% higher values of annual residential NOx emissions in GEAA and TCNA (Figure 9d) with respect to EDGAR is produced by a higher emissions factor adopted in Argentina (TCNA) for NG emissions (150 g/GJ) compared to 51 g/GJ proposed by EMEP (EMEP2019 Table 3.3 section 1.A.4.b.i). Have we adopted 51 g/GJ as from EMEP, then we would have obtained a lower total annual NOx emissions, consistent with less primary energy use (firewood, others).*

R1: Line 654: Here emissions in sector 1B1 (fugitive from solid fuel production) are mentioned, but I cannot find any description in the Methods section on how these are estimated. I believe EDGAR uses a constant emission factor per produced tons of coal. Do the authors use a different method? Do they have further comments on this? I think EDGAR's fugitive emissions in general are very approximate, and any pointers on how this could be improved would surely be welcomed.

*A: Thanks for the suggesting. We will add the following discussion in the Methods section of the revised manuscript. We have checked the coal production from national bases and the information in the Argentine National Energy Balance-NEB- (Figure A6), and the IEA data bases (Figures A7-A9). Comparing the national records and the IEA, both records are consistent. In GEAA, we have calculated the 1B1 sector (solid fuel production) using only the national production, estimated from the Argentine NEB, which is also used in TCNA. We applied two emission factors for mining and post-mining operations: 18 $m^3$ $CH_4$/t and 2.5 $m^3$ $CH_4$/t gross production of coal, respectively (IPCC Chap 4). Retro-calculating the coal amount used in EDGAR (computing the amount of coal from $CH_4$ emissions) assuming the same IPCC emissions factors, we obtain the black line in Figure A6. Therefore, we can assume that EDGAR has used a percentage of total coal uses (net production + import) or used another emission factor. This overestimation produces an annual average emission difference of 22.3 kt of $CH_4$ between EDGAR and GEAA.*

[Figure]

***Figure A6:*** *Coal production and coal import in Argentina (ktoe). Red line data used in GEAA (only national coal gross production), green line: NEB data of import + national net production, black: estimation of EDGAR data, blue line net coal production.*

[Figure]

*Figure A7:* Coal production in Argentina (ktoe) IAE. Source: https://www.iea.org/data-and-statistics/data-browser?country=ARGENTINA&fuel=Coal&indicator=CoalConsByType

[Figure]

*Figure A8:* Coal imports vs export in Argentina (ktoe) IAE. Source: https://www.iea.org/data-and-statistics/data-browser?country=ARGENTINA&fuel=Coal&indicator=CoalConsByType

[Figure]

IEA. All rights reserved.

*Figure A9:* *final consumption by type in Argentina (ktoe) IAE. Source: https://www.iea.org/data-and-statistics/data-browser?country=ARGENTINA&fuel=Coal&indicator=CoalConsByType*

*A: In the methodological section "2.3.2 Fuel production sector" (lines 200-205) former manuscript says:*
*"Emissions from the production and transformation of fuels were calculated from own consumption, venting, and flaring in refineries, and the production from oil and gas in wells. The Ministry of Energy (Minem, 2020) maintains a monthly record of up-stream (production and extraction of gas and oil) in the wells and down-stream (fuel production, own consumption, and sales) in the refineries. Emissions were calculated from own consumption (in wells and refineries) according to the type of fuel consumed, using Eq. (1). In a GIS format, each well or refinery are represented as point sources, so the emissions are in their respective coordinate."*

*we will change by:*

*"Emissions from the production and transformation of fuels were calculated from own consumption, venting, and flaring in refineries, and the production from oil and gas in wells. Within the solid fuel production sector (1B1) we estimated the gross production of coal using the Argentine National Energy Balance-NEB. We applied two emission factors for mining and post-mining operation (18 $m^3$ $CH_4$/t and 2.5 $m^3$ $CH_4$/t gross production of coal, respectively, IPCC Chap 4) based on mining activity in Río Turbio, Santa Cruz (-51.57, -72.31). The Ministry of Energy (Minem, 2020) maintains a monthly record of up-stream (production and extraction of gas and oil) and down-stream (fuel production, own consumption, and sales) activities in wells and refineries. Emissions were calculated from own consumption (in wells and refineries) according to the type of fuel consumed, using Eq. (1). In a GIS format, each well or refinery are represented as point sources, so total emissions are located at their respective coordinate."*

*Lines 654-658 former manuscript says:*

*"On the other hand, for the fuel production and fugitive emissions subsectors (1A1cb, 1B1 and 1B2), GEAA-AEIv3.0M has an important difference with respect to EDGAR, especially in methane emissions being EDGAR more than 90 % larger than GEAA (for the sum of subsectors). These differences totalize 598 Gg of $CH_4$ (or 14,970 Gg $CO_{2eq}$) per year (Figure 7 and Table 8 App.). Note that for the particular case of the 1B1 sector (fugitive emissions from coal mining), the activity data for the GEAA inventory has been estimated from the national primary energy balance, which possess large uncertainties (TCNA, 2015)"*

*Lines 654-658 will be changed by:*

*On the other hand, for the fuel production and fugitive emissions subsectors (1A1cb, 1B1 and 1B2), GEAA-AEIv3.0M has an important difference with respect to EDGAR, especially in methane emissions: EDGAR annual $CH_4$ emissions are more than 90 % larger than GEAA (for the sum of subsectors). These differences totalize 598 Gg of $CH_4$ (or 14,970 Gg $CO_{2eq}$) per year (Figure 7 and Table 8 App.). Note that for the particular case of the 1B1 sector (fugitive emissions from coal mining), the activity data for the GEAA inventory has been estimated from the national primary energy balance, which possess large uncertainties (TCNA, 2015). As mentioned above, although EDGAR uses the Energy Balances from IEA, which in turn is based on national energy balances, the amount of coal computed from $CH_4$ emissions in EDGAR, using the same IPCC emissions factor, seems to be proportional to the total coal uses (net production + import of coal; see Figure S18, Suppl mat).*

[Figure]

**_Figure S18:_** *Coal production and coal import in Argentina (ktoe). Red line data used in GEAA (only national coal gross production), green line: National Energy Balance (NEB) data of import + national net production; blue line NEB net coal production; black line: estimation of EDGAR data.*

R1: Lines 659ff: With respect to differences in N2O emissions, the point is made that this could be inclusion/exclusion of LULUCF N2O emissions. Do the authors know whether the EDGAR grids include LULUCF N2O emissions? Could a comparison additionally be made to EDGAR non-gridded data, which I believe do allow exclusion of LULUCF emissions?

A. *EDGAR $N_2O$ temporal series 1970-2015 for Argentina in the Manure management and Agriculture sectors (3A-3C) presents the following subsectors: 3.A.2: Manure Management, 3.C.1: Emissions from biomass burning, 3.C.4: Direct $N_2O$ Emissions from managed soils, 3.C.5: Indirect $N_2O$ Emissions from managed soils, 3.C.6: Indirect $N_2O$ Emissions from manure management. The recent new gridded maps EDGAR 6.0 (v. 2018) includes subsectors 3C1b Agricultural waste burning; (3C2+3C3+3C4+3C7) Agricultural soils 3C5+3C6; 4D3 Indirect $N_2O$ emissions from agriculture.*
*The focus of the current paper is producing a pollutant map for the air quality modelling, which besides the important climate impacts of $N_2O$ source strength does not directly influence on local air quality. Thus, a comparison with respect to the new version of EDGAR (6.0) is out of the scope of for the present work, and will probably be an activity for a future research. We have presented gridded comparison of $PM_{10}$ and NOx in Figures 8 and 9 and similar figures can be produced for other pollutants*

R1: Finally, some comment on how readily the dataset might be updated in future would be of interest.

A: *We are now updating the inventory to April 2021 with the latest available information.*

---

## Author Comment (AC2)

**Response to referee comments 3 (RC3)**

Dear Referee 3,

Thank you for the revision of our manuscript number ESSD-2021-81. We have reviewed and adjusted the manuscript considering all your observations and commentaries. Below, you will find our detailed responses. Within this response letter the following style is used: the original general comments made by the reviewer are kept in normal text (initiated with R), our responses are *in blue italics initiating with A* (Authors). We will use *italics black* for other authors texts (citations, initiated with C). The corresponding edit in the manuscript will be included in red. In addition, we attach the appendix section below with the new suggested changes highlighted in yellow.

**Referee comments 3 (RC3)**

Comment on essd-2021-81 Anonymous Referee #3. Referee comment on "High resolution seasonal and decadal inventory of anthropic gas phase and particle emissions for Argentina" by S. Enrique Puliafito et al., Earth Syst. Sci. Data Discuss., https://doi.org/10.5194/essd-2021-81-RC3, 2021

R3. Emission inventories are a critical input to air quality and climate models, while we lack comprehensive regional emission inventories over Argentina for a long time. The authors have developed an anthropogenic emission inventory for Argentina from 1995 to 2020, which is of great importance for the scientific community. The local activity database and emission factors used in this work improve the estimates of anthropogenic emissions in Argentina compared to global emission inventories. Overall, I think that this study provides important and useful emission datasets and is publishable in the journal of ESSD.

*A. Thank you very much for your positive an encouraging comment.*

R3. My only concern is that the uncertainty of the estimated emissions is not quantitatively assessed with the uncertainty range, and the comparison with global emission inventories lacks the CEDS inventory, which should be included in the analysis.

*A. Thank you very much for the interesting and constructive suggestion. Besides the already presented comparison with EDGAR data base and TCNA 2015 (Argentine inventory) we will included in the revised manuscript, as suggested, a comparison with CEDS international database for several individual sectors and pollutants in the form of total annual time series from 1995 to 2015. In doing so, we will maintain the comparison with respect to EDGAR, as already done in the original manuscript. It must be noted that, according to Hoesly et al, (2018) and McDuffie et al, (2020), compilers of CEDS database, for Argentina they have used the TCNA 2015 Argentine inventory, so, in some senses the suggested comparison was already presented in the initial manuscript. Nevertheless, we will explicitly include CEDS in each respective section and add a supplementary material with the full annual comparison among the inventories.*
*We must also add an additional comment concerning the Argentine inventory. Argentina has presented the third biennial update to UNFCC in 2019. The official data posted in the governmental page (https://inventariogei.ambiente.gob.ar/resultados last access July 27, 2021) has some differences with the previous TCNA, 2015, so, we will include both inventories for Argentina which we will be calling TCNA2015 (1990-2012) and TCNA2019 (1990-2016).*

Based on Table 1b of the original text, the comparisons include the following sectors and pollutants:

| Sector and Activities | $CO_2$ | $CH_4$ | $N_2O$ | CO | NOx | $SO_2$ | NMVOC | TSP | PM10 | PM2.5 | BC |
|---|---|---|---|---|---|---|---|---|---|---|---|
| **Fuel Combustion:** | | | | | | | | | | | |
| Power and heat production | abcde | abcde | abcde | abcde | abcde | abcde | abcde | ae | ae | ae | ae |
| Fuel Production (incl. fugitive emissions, venting, and flaring) | abcde | abcde | abcde | abcde | abcde | abcde | abcde | ae | ae | ae | ae |
| Road transportation | abcde | abcde | abcde | abcde | abcde | abcde | abcde | ae | ae | ae | ae |
| Domestic aviation | abcde | abcde | abcde | abcde | abcde | abcde | abcde | ae | ae | ae | ae |
| Railroad and navigation | abcde | abcde | abcde | abcde | abcde | abcde | abcde | ae | ae | ae | ae |
| Residential Commercial and Public offices combustion | abcde | abcde | abcde | abcde | abcde | abcde | abcde | ae | ae | ae | ae |
| Fuel use in agriculture / others | abcde | abcde | abcde | abcde | abcde | abcde | abcde | ae | ae | ae | ae |
| **Industrial Processes (non-combustion):** | | | | | | | | | | | |
| Production of minerals, chemicals, and metals, pulp/paper/food/drink | abcde | abcde | abcde | ade | ade | ade | ade | ae | ae | ae | ae |

a. GEAA (1995-2015); b. TCNA2015 (1995-2012); c: TCNA2019 (1995-2014); d: CEDS (1995-2014); e: EDGAR (1995-2015)

The explicit comparison in form of figures and tables is organized as a supplementary file "comp_geaa_ceds_edgar_tcna.xlsx", which contains detailed annual temporal profile information for each inventory. It includes tables and figures according to the following index:

**Table A6: Index of supplementary file:** "comp_geaa_ceds_edgar_tcna.xlsx"

Page 1    Summary table for all species and sectors
Page 2    Summary tables for $CO_2$ all sectors and inventories
Page 3    Tables and Figures for $CO_2$ all sectors and inventories
Page 4    Summary tables for $CH_4$ all sectors and inventories
Page 5    Tables and Figures for $CH_4$ all sectors and inventories
Page 6    Summary tables for $N_2O$ all sectors and inventories
Page 7    Tables and Figures for $N_2O$ all sectors and inventories
Page 8    Summary tables for CO all sectors and inventories
Page 9    Tables and Figures for CO all sectors and inventories
Page 10    Summary tables for $NO_X$ all sectors and inventories
Page 11    Tables and Figures for $NO_X$ all sectors and inventories
Page 12    Summary tables for NMVOC all sectors and inventories
Page 13    Tables and Figures for NMVOC all sectors and inventories
Page 14    Summary tables for $SO_2$ all sectors and inventories
Page 15    Tables and Figures for $SO_2$ all sectors and inventories
Page 16    Summary tables for $NH_3$ all sectors and inventories
Page 17    Tables and Figures for $NH_3$ all sectors and inventories

This index will be explicitly included in the Appendix of the manuscript. Also Tables A7 through A10 (from the Appendix) summarizes the main results of the inter-comparison study. The main results are presented in table A7, which we copy here:

[revised manuscript text omitted]

Ref.: mad: Mean absolute differences between GEAA-AEIv3.0M and the other captioned inventory for years 1995-2015. sd.: Standard deviation of the two inventories for years 1995-2015. AVERAGE includes the mean values of TCNA2015 (1995-2014) and CEDS (1995-2014).

To include the new explicit comparison in the main text, we will modify several sections in the manuscript to introduce the comparison with CEDS:

In the Abstract section says:

"Spatial and temporal comparisons were also performed against EDGAR HTAPv5.0 inventory for several pollutants. The agreement was acceptable within less than 30% for most of the pollutants and activities, although a >90% discrepancy was obtained for methane from fuel production and fugitive emissions and >120% for biomass burning".

It changes:

"Temporal comparisons for several pollutants were also performed against two international databases: Community Emissions Data System (CEDS) and EDGAR HTAPv5.0 inventories; for EDGAR it also includes a spatial comparison. The agreement was acceptable within less than 30% for most of the pollutants and activities, although >90% discrepancy was obtained for methane from fuel production and fugitive emissions and >120% for biomass burning"

In the Introduction section, Lines 117… says:

"We compare our results with the Argentine GHG inventory for the Third National Communication of Argentina to the IPCC (TCNA, 2015), which includes annual GHG emissions from 1990 through 2014. Annual and monthly emissions of air quality pollutant such as PM and NOx are also compared to the estimations presented in the EDGAR HTAPv5.0 inventory (Crippa et al., 2016, 2020; EDGAR, 2019)"

Which changes to:

"We compare our results with the Argentine GHG inventory for the Third National Communication of Argentina to the IPCC (TCNA, 2015), which includes annual GHG emissions from 1990 through 2014, and was further updated in 2019 (TCNA, 2019), spanning from years 1990 to 2016. Annual total emissions of GHG and air quality pollutants are also compared to the estimations presented in the EDGAR HTAPv5.0 inventory (Crippa et al., 2016, 2020; EDGAR, 2019) and Community Emissions Data System (CEDS) (Hoesly, et al. 2018; McDuffie et al, et al, 2020)"

In Section 4. Inter-comparison of GEAA-AEIv3.0M with other Emissions Inventories for Argentina (Lines 582…). It says:

"Since the present GEAA-AEIv3.0M inventory includes spatial and temporal variation, its calibration requires a double control and validation. For the temporal comparison we use the Argentina national greenhouse gas inventory (TCNA, 2015) that compiled the total annual values for Argentina between 1990 and 2014"

Which changes as:

"Since the present GEAA-AEIv3.0M inventory includes spatial and temporal variation, its calibration requires a double control and validation. For the temporal comparison we use the Argentina national greenhouse gas inventory (TCNA, 2015) that compiled the total annual values for Argentina between 1990 and 2014 and an updated version in 2019 (TCNA, 2019) spanning from years 1990 to 2016. In addition the most commonly used international inventories EDGAR HTAPv5.0 and CEDS are also considered. It should be noted that CEDS uses TCNA 2015 as a basis for the Argentine information (Hoesly et al, 2018), but for some species and sectors they differ. There are also some differences between TCNA 2015 and TCNA 2019 prior to year 2014. Therefore, we will compare GEAA with 4 temporal series: TCNA2019, TCNA2015, CEDS and EDGAR"

In the following lines (585…) says:

*"Although the activity data for both studies were taken basically from the same national sources, the focus and methodology of each inventory varies. In TCNA activities and emissions are accumulated using a top-down approach to obtain a nation-wide annual total by sector. While in our case (GEAA-AEIv3.0M) the activities and emissions are first located in each point, line, or area with a bottom-up approach, and then the totals are calculated as the sum of all cells in the spatial grid. Therefore, the sum of the activities by sector and year may vary slightly.*

*Likewise, we compare the annual values with the international EDGAR inventory, which differs especially in the use of proxy variables used for its spatial disaggregation, which has already been discussed elsewhere (Puliafito et al., 2015, 2017). A spatial comparison can also be made with the EDGAR inventory, although it has a resolution of 0.1° × 0.1°, which requires an adaptation of our higher resolution inventory (0.025° × 0.025°)."*

Now changes as:

*"Although the activity data for GEAA and TCNA (and therefore CEDS) were taken basically from the same national sources (mostly from the National Energy Balance), the focus and methodology of each inventory varies. In TCNA activities and emissions are accumulated using a top-down approach to obtain a nation-wide annual total by sector. While in our case (GEAA-AEIv3.0M) the activities and emissions are first located in each point, line, or area with a bottom-up approach, and then the totals are calculated as the sum of all cells in the spatial grid. Therefore, the sum of the activities by sector and year may vary slightly. With respect to EDGAR, it differs in the use of proxy variables for its spatial disaggregation, which has already been discussed elsewhere (Puliafito et al., 2015, 2017). A spatial comparison with the EDGAR inventory is presented in section 4.2"*

In Section 4.1 (lines 604…) says
*" 4.1 Comparison with total annual values from TCNA"*

And Lines 607…
*"Figure 7a shows the annual values for both inventories, and Figure 7b shows the average annual differences by activity Table A7 (App.). Most of the activities (1A1, 1A2, 1A1b, 1A1c, 1A3a, 1A3b, 1A4a-b, 2B, 2C, 3A, 3C, see Table 1a) agree within ± 6.0 % with total differences for the sum of all sectors of 0.4 ± 3.9 %. Higher discrepancies are found in sector 1A1c (FPR 7%), 1A3c-d (R+N: 13.3%), 3C (AG: -12.5%) and (AWB -6.5%). For fuel production, the discrepancy arises from the way the activity is computed".*

Will change as:

*" 4.1 Comparison with total annual values from TCNA, EDGAR and CEDS"*

*"Figure 7a shows the annual values for TCNA2019, TCNA2015, CEDS and EDGAR inventories, and Figure 7b shows the average annual differences by activity Table A7 (App.). Most of the activities (1A1, 1A2, 1A1bc, 1B1, 1B2, 1A3a, 1A3b, 1A4abc, 2B, 2C, 3A, 3B, see Table 1a) agree within ± 16.0 %. Higher discrepancies are found for $N_2O$ and $CH_4$, and in sectors 1B1 (FPR >100%), 1B2 (FUG>50%), 1A3c-d (R+N: 13.3%), 3C (AG: -12.5%) and (AWB -6.5%). For fuel production, the discrepancy arises from the way the activity is computed.*

(see also discussion below in manuscript and response to Reviewer 1 on solid fuel emissions discrepancies with EDGAR)

Figure 7 changes to:

a)

[Figure]

b)

[revised manuscript text omitted]

R2. Besides, I would like to see the evaluation of emission trends with top-down observational constraints, such as comparing NOx and SO2 emissions estimated in this study with NO2 and SO2 retrievals from the OMI satellite. There are also some top-down inversion products of global CO, NOx, and SO2 emissions available at present, which can be used to extract and summarize the emissions over Argentina and evaluate the bottom-up emission inventory developed in this study

A.  Thank you for the interesting suggestion. It would be very interesting indeed to include these satellite comparisons, but we feel that this is out of the scope of this compiled description and inter-comparison of the GEAA inventory. Including the satellite comparison would require adding a new methodological section presenting the satellite instruments and measurements, the retrieval algorithm, its uncertainties, and so on. We estimate that such comparisons study could be a paper by its own, and Reviewer R1 already highlighted the importance of bringing together the work undertaken over several years into a single comprehensive study covering many sectors and a range of emission species. Moreover, comparing with tropospheric column (i.e., using OMI or TROPOMI) requires a full atmospheric model like WRF-Chem, which we have only recently implemented in our group. On the other hand, the time given of 4 weeks is scarce to prepare this new research. Reviewing other inventories papers for example those presented by the EDGAR team (i.e., Crippa et al, 2020; in Janssens-Maenhout, G.et al, 2019; or by the CEDS team, the above-mentioned McDuffie et al, 2020 or Hoesly et al, 2018; and many others) do not include a satellite retrieval validation in the presentation of their inventory paper. For example, Fioletov et al, 2011, use several years of OMI measurements to calibrate the retrievals for SO$_2$ in the US, but does not describe in detail the emission inventory used. In summary, as much as we would like to present such study in this paper, we do not have time to do it now, and most importantly, we will rather focus on a future manuscript specifically centered at performing a spatio-temporal comparison with respect to satellite retrievals.

Boersma, K. F., Eskes, H. J., Dirksen, R. J., van der A, R. J., Veefkind, J. P., Stammes, P., Huijnen, V., Kleipool, Q. L., Sneep, M., Claas, J., Leitão, J., Richter, A., Zhou, Y., and Brunner, D.: An improved tropospheric NO$_2$ column retrieval algorithm for the Ozone Monitoring Instrument, Atmos. Meas. Tech., 4, 1905–1928, https://doi.org/10.5194/amt-4-1905-2011, 2011.

V. E. Fioletov, C. A. McLinden, N. Krotkov, M. D. Moran, and K. Yang. Estimation of SO2 emissions using OMI retrievals; Geophysical Research Letters, VOL. 38, L21811, doi:10.1029/2011GL049402, 2011

[revised manuscript text omitted]